# Δ133p53β isoform pro-invasive activity is regulated through an aggregation-dependent mechanism in cancer cells

Nikola Arsic[1,2], Tania Slatter [3], Gilles Gadea[4], Etienne Villain [1,2], Aurelie Fournet[5,2], Marina Kazantseva [3], Frédéric Allemand[5,2], Nathalie Sibille[5,2], Martial Seveno [6,2], Sylvain de Rossi[7], Sunali Mehta[3], Serge Urbach [8,2], Jean-Christophe Bourdon [9], Pau Bernado[5,2], Andrey V. Kajava [1,2,10], Antony Braithwaite [3] & Pierre Roux [1,2✉]

The p53 isoform, Δ133p53β, is critical in promoting cancer. Here we report that Δ133p53β activity is regulated through an aggregation-dependent mechanism. Δ133p53β aggregates were observed in cancer cells and tumour biopsies. The Δ133p53β aggregation depends on association with interacting partners including p63 family members or the CCT chaperone complex. Depletion of the CCT complex promotes accumulation of Δ133p53β aggregates and loss of Δ133p53β dependent cancer cell invasion. In contrast, association with p63 family members recruits Δ133p53β from aggregates increasing its intracellular mobility. Our study reveals novel mechanisms of cancer progression for p53 isoforms which are regulated through sequestration in aggregates and recruitment upon association with specific partners like p63 isoforms or CCT chaperone complex, that critically influence cancer cell features like EMT, migration and invasion.

[1] Université de Montpellier, Centre de Recherche en Biologie Cellulaire de Montpellier (CRBM) CNRS, UMR 5237, Montpellier, France. [2] Université de Montpellier, Montpellier, France. [3] Department of Pathology, University of Otago, Dunedin, New Zealand. [4] Université de la Réunion, Unité Mixte 134 Processus Infectieux en Milieu Insulaire Tropical, INSERM Unité 1187, CNRS Unité Mixte de Recherche 9192, IRD Unité Mixte de Recherche 249. Plateforme Technologique CYROI, Sainte Clotilde, France. [5] Centre de Biochimie Structurale (CBS), INSERM, CNRS, 29 rue de Navacelles, Montpellier, France. [6] BioCampus Montpellier, CNRS, INSERM, Montpellier, France. [7] MRI, UMS BioCampus Montpellier, CNRS, INSERM, Université de Montpellier, Montpellier, France. [8] IGF, CNRS, INSERM, Montpellier, France. [9] Dundee Cancer Centre, University of Dundee, Ninewells Hospital and Medical School, Dundee, UK. [10] Institut de Biologie Computationnelle, Montpellier, France. ✉email: pierre.roux@crbm.cnrs.fr

nactivation of p53 is a common event in cancer progression. About 50% of human cancers display mutations in *TP53*, while in the remaining cases the p53 pathway is inactivated by other mechanisms[1,2]. Upon mutation p53 not only stops to provide anti-oncogenic activities but it becomes a very strong tumour inducer. Mutated p53 is not able to maintain genome stability due to the absence of cell cycle and pro-apoptosis regulation[3,4]. In addition, p53 mutants strongly induce cancer cell migration and invasion[5,6].

Recently it was demonstrated that mutated and particularly structural p53 mutants could create protein aggregates[7] with amyloid characteristics[8,9]. Interestingly co-aggregation of mutant p53 with some of its interacting partners like p63 and p73 was suggested as mechanism for its gain of function activity that induces cancer progression[10]. In addition, correct folding of p53 by the chaperone complex CCT (Chaperonin Containing TCP1 or TriC-TCP-1 Ring Complex) was demonstrated to be crucial for its functions in migration and invasion[11].

The *TP53* gene encodes 12 different isoforms generated by alternative promoter usage, alternative splicing and alternative transcription starting site usage (reviewed in ref. [12]). p53 isoforms are organised into four major groups containing three members for each: TAp53 [α, β, and γ], Δ40p53 [α, β and γ], Δ133p53 [α, β and γ] and Δ160p53 [α, β, and γ]. p53 isoforms are modified versions of the canonical p53 protein (TAp53α-TA stands for "transactivation"), either truncated on N-terminus (Δ40, Δ133 and Δ160) or having different C-terminal oligodimerisation domains (α, β or γ). An increasing body of literature demonstrates that p53 isoforms are involved in the regulation of cell-cycle progression, programmed cell death, replicative senescence, cell differentiation, viral replication, and angiogenesis[13–18] and that some of them are deregulated in human tumours[19]. For example, overexpression of Δ133p53 in mouse models was reported to be involved in tumour formation through persistent inflammation in an IL-6 dependent manner[20,21] as well as in the promotion of the tumour invasion through activation of JAK-STAT and RhoA-ROCK signalling pathways[20]. Aberrant expression of Δ133p53 isoforms and Δ133p53β particularly was demonstrated to promote metastatic spread[22] which correlates with increased risk of recurrence in breast, colorectal, glioblastomas and prostate cancers, independent of *TP53* mutations[20,22–25]. In addition, wild type (WT) Δ133p53β was demonstrated to promote cancer stem cell potential[26]. Thus, WT Δ133p53β appears to have intrinsic oncogenic activities, functioning in opposition to the tumour suppressive functions of canonical p53 protein. Thus cancer-promoting activities of WT Δ133p53β isoform are functionally very similar to those of mutated p53 proteins.

Although a body of literature shows critical roles of p53 isoforms and particularly Δ133p53β in cancer progression, the regulation of its activity is not well understood. Here we show that Δ133p53β activity is regulated through an aggregation-dependent mechanism. The aggregation capacity of WT Δ133p53β was evaluated using biochemical, cellular and computational approaches as well as by in vitro analysis using recombinant protein. Aggregates were also confirmed in vivo by immunohistochemical detection in tissues from the Δ122p53 transgenic mouse model of Δ133p53[27] and also in human tumours.

WT Δ133p53β aggregation capacity inversely correlates with its activity in cancer cell invasion and migration. Furthermore, WT Δ133p53β aggregation depends on binding to specific regulatory partners. Depletion of CCT chaperone complex, that associates with WT Δ133p53β, increases aggregation and abolishes the pro-migratory effect of this isoform. In contrast, association with p63 family members reduces aggregation of WT Δ133p53β, and

significantly modulates the invasive capacity of breast cancer cell lines.

Taken together, our data provide evidence for a regulatory mechanism of Δ133p53β activity through its aggregation capacity. Our data suggest that the balance between aggregated and non-aggregated Δ133p53β proteins determine the level of its activity in cancer cells. Interactions with partners that promote Δ133p53β activity, like CCT subunits or p63 family members thus offer interesting new therapeutic opportunities for treatment of metastatic disease in cancers expressing this isoform.

## Results

**The WT Δ133p53β creates cellular aggregates**. A growing body of literature has demonstrated functional similarities between the small p53 isoforms, particularly WT Δ133p53β, and mutated p53 in cancer progression[21,22,24,26,28]. In addition, it was shown recently that structurally mutated p53 proteins are able to create protein aggregates in cancer cells. Here we asked whether WT Δ133p53β also has the capacity to create protein aggregates in cancer cells.

To do this a multidisciplinary approach was adopted. We first used a modified western blot assay. As aggregates are more resistant to SDS and heat denaturation than non-aggregating proteins, extracts were heated at 42 °C or 95 °C and the aggregates visualized by an upward shift in the molecular mass on SDS-PAGE, consistent with the formation of large multimeric assemblies. In addition, immunofluorescence analysis was performed for direct visualisation of aggregates upon confocal microscopy followed by 3D reconstruction of Z-stacks. We then studied the presence of the Δ133p53β aggregates in a mouse transgenic model and in human tumours. Finally, we evaluated the aggregation capacities of recombinant WT Δ133p53β protein.

In order to evaluate aggregate forming capacity of p53 and its mutants, plasmids expressing WT p53, p53R175H or p53R273H as well as plasmids expressing WT and mutated Δ133p53β isoforms, namely WT Δ133p53β, Δ133p53βR175H and Δ133p53βR273H were transfected into lung cancer H1299 cell line, which is devoid of endogenous p53, thus convenient for the study of individual isoform characteristics. We found structural p53R175H, but not WT p53 created aggregates at 42 °C. Contact mutant p53R273H also exhibited some very low level of aggregation forming capacity (Fig. 1a). These observations are in agreement with literature showing that p53 mutations (including structural mutants like R175H) affect the correct folding of the protein leading to the 'denaturation' of the p53 protein, which is pre-requisite for aggregation. On the other hand, mutations that affect the DNA-binding domain (contact mutants like R273H), do not affect folding and thus do not contribute much to the aggregation[29].

In addition, Δ133p53β isoform and its mutants were all able to form pronounced aggregates, which surprisingly, appear to be more resistant to heat denaturation at 95 °C compared to those of p53R175H (Fig. 1a). To complete the biochemical analysis, we also examined the non-soluble cellular fractions. The protein extraction procedure enables us to separate soluble proteins (annotated as "Supernatant" in figures) from cellular debris and the non-soluble protein fraction (annotated as "Pellet" in figures). While small quantities of p53R175H structural mutant were detected among non-soluble proteins a significant fraction of Δ133p53β was observed in the non-soluble protein fraction (Fig. 1b).

Immunofluorescence analysis revealed that only structural p53R175H mutant created visible dot-like aggregates partially co-localising with amyloid-type aggregates, as detected with OC (α-Aβ42) antibody that specifically detects fibril aggregates (Fig. 1c

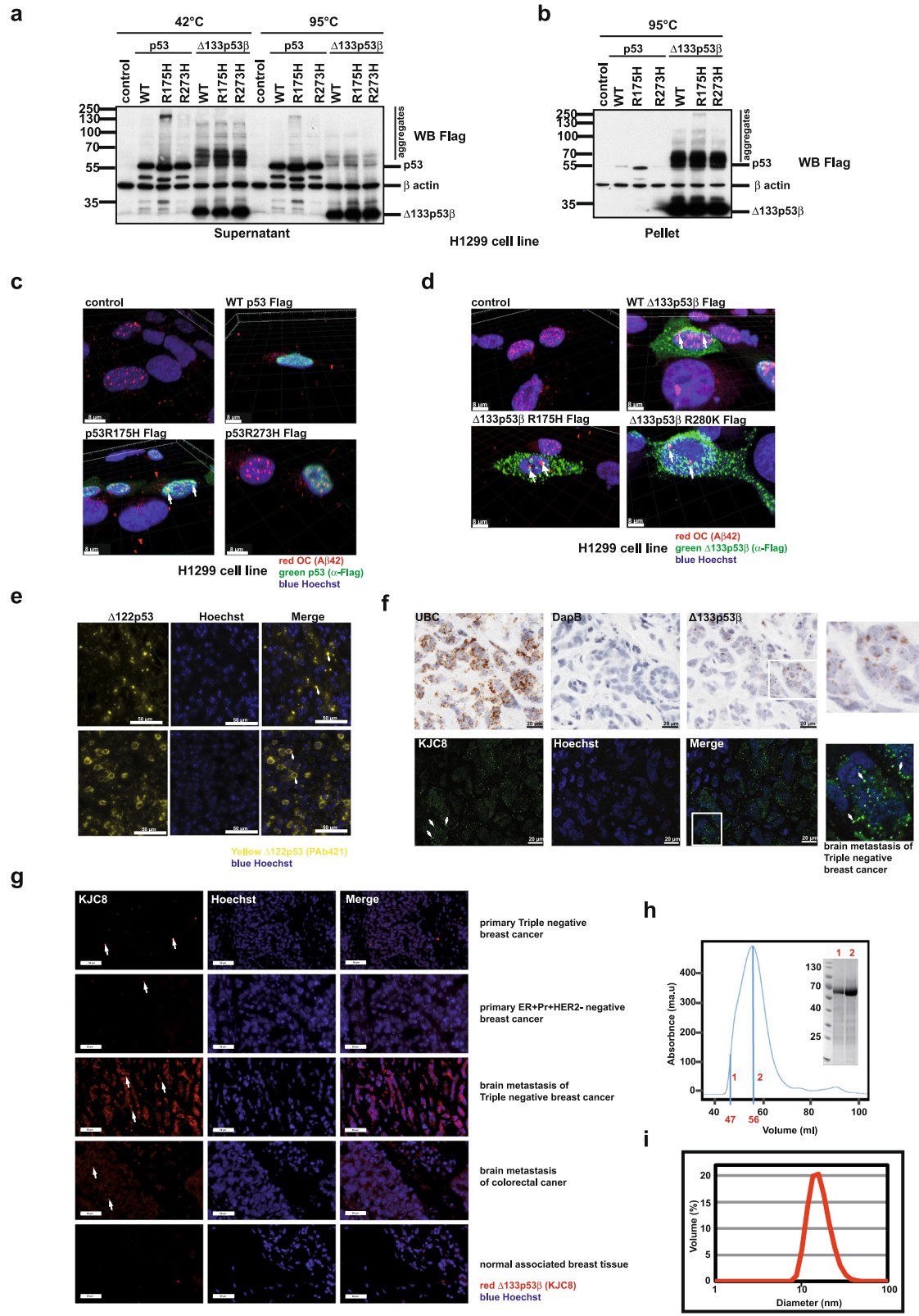

and Supplementary Fig. 1a). This is in agreement with the results obtained by western blot analysis and previously reported data[8]. Conversely, WT and mutant R273H p53 proteins showed diffused nuclear staining, supporting previous finding[10] and almost no colocalisation with OC (α-Aβ42) labelled amyloid-type aggregates. However, a marked punctate staining revealed Δ133p53β aggregates predominantly in cytoplasm for both WT

and mutant proteins. Interestingly Δ133p53β aggregates are more pronounced than those formed from p53R175H mutant. Similarly to p53R175H aggregates, Δ133p53β only partially co-localized with amyloid-type aggregates detected by OC (α-Aβ42) antibody (Fig. 1c and Supplementary Fig. 1b). In addition, p53R175H exhibited partial co-localisation with oligomers detected by A11 antibody that recognizes amino acid

**Fig. 1 Evaluation of the aggregate forming ability of p53 and Δ133p53β. a** Western blot analysis of the soluble protein fraction from H1299 cells after transfection with WT, structurally and contact mutated p53 and Δ133p53β. $n = 3$. **b** Western blot analysis of the insoluble protein fraction from H1299 cells after transfection with WT, structurally and contact mutated p53 and Δ133p53β. $n = 3$. **c** 3D reconstruction of confocal immunofluorescent Z-stack images of H1299 cells transfected with WT, structurally and contact mutated p53. Green: α-Flag (p53), red: α-OC (Aβ42, amyloid aggregates) antibody, blue: Hoechst. Scale bar 8 µm. $n = 3$. **d** 3D reconstruction of the confocal immunofluorescent Z-stack images of H1299 cells transfected with WT, structurally and contact mutated Δ133p53β. Green: α-Flag (Δ133p53β), red: α-OC (Aβ42, amyloid aggregates) antibody, blue: Hoechst. Scale bar 8 µm. $n = 3$. **e** Immunofluorescence analysis of p53 (Δ122p53) localisation in tissues obtained from Δ122p53 homozygote mice performed using the PAb421 antibody. Yellow: p53 (Δ122p53), Blue: Hoechst. In total ten individual tumours from the Δ122p53 animal model were analysed. Arrows in upper panel indicate prominent cytoplasmic aggregates while in lower panel arrow demonstrates aggregates in between diffuse cytoplasmic staining. Scale bar 50 µm. **f** Upper panel: RNAscope to identify metastatic triple-negative breast tumours expressing Δ133p53β. Ubiquitin C (UBC) as a positive and the bacterial gene DapB as a negative control. Lower panel: Immunofluorescence analysis using the KJC8 antibody to detect Δ133p53β aggregates in a metastatic triple-negative breast tumour. Boxed areas: digital magnifications of regions with characteristic staining. $n = 5$. Scale bar 20 µm. **g** Immunohistochemical analysis of Δ133p53β aggregate presence in different primary or metastatic tumours as well as in normal breast tissue. Arrows indicate prominent Δ133p53β aggregates. Scale bar 50 µm. $n = 5$ (for primary breast cancers and corresponding brain metastases), n = 15 for normal associated breast tissue and $n = 3$ for brain metastasis of colon cancer. **h** Solution studies of the MBP-Δ133p53β construct. Size exclusion chromatography (SEC) profile of recombinant MBP-Δ133p53β injected on a Superdex 200 16/60. The maximum of the peak corresponds to an elution volume at 56 mL, which is close to the dead volume for that column. The size of the protein is estimated at 440 kDa according to the calibration. SDS-PAGE of SEC samples collected at elution times of 47 ml (line 1) and 56 ml (line 2). The observed SDS-PAGE bands are in agreement with the expected molecular weight of MBP-Δ133p53β (67.4 KDa). **i**. Dynamic light scattering (DLS) profile of the maximum peak from the SEC (2). The profile displays a unique peak with a maximum corresponding to a particle with a diameter of 16.7 nm and an estimated molecular weight of 480 KDa. The peak is broad and most probably corresponds to the equilibrium of multiple species of variable sizes. $n = 3$. The arrows indicate co-localisation in Fig. 1c and d or aggregate Fig. 1e–g. Molecular weight is expressed in kDa. Source data are provided as a Source Data file.

sequence-independent oligomers of proteins or peptides. Such co-localisation was hardly observed for Δ133p53β protein suggesting that mutant p53 and Δ133p53β are structurally different (Supplementary Fig. 1c, d).

We also evaluated the aggregate forming capacity of WT and mutant Δ133p53β isoforms in breast cancer cell models. This cancer is well stratified into sub-types each possessing different degrees of invasiveness: triple negative (represented by strongly invasive MDA-MB-231), luminal B (represented by moderately invasive MCF7), as well as different mutational status of p53, either wild type (MCF7), structural mutant (p53R175H in SK-BR-3), or contact mutant (p53R280K in MDA-MB-231). Different breast cancer cell models allow investigation of the interplay between p53 mutational status, cancer aggressiveness and Δ133p53β aggregation.

WT Δ133p53β was transfected into MCF-7, Δ133p53βR280K in MDA-MB-231 D3HLN and Δ133p53βR175H into the SK-BR-3 cell line. As shown in Supplementary Fig. 2a, aggregates were detected in all cell lines. Similar to the results from H1299 cells, WT and R280K mutant of Δ133p53β in MCF-7 and MDA-MB-231 D3HLN respectively displayed a distinct "dotty" pattern with partial co-localization with amyloid type aggregates. The SK-BR-3 cell line however showed many and large amyloid aggregates, which were co-localized with Δ133p53βR175H isoform. Large numbers of aggregates in this cell line are not surprising since the SK-BR-3 cell line also expresses the p53R175H protein. Finally, we further demonstrated the presence of aggregates in each cell line using western blot analysis (Supplementary Fig. 2b).

Next, we analysed the presence of aggregate expression pattern of Δ133p53β protein in a mouse model expressing an N-terminal deletion mutant of p53 (Δ122p53, a mouse model of Δ133p53),[27],[30]. Δ122p53 mice were shown to have decreased survival and aggressive tumour spectrum, marked proliferative advantage and reduced apoptosis, as well as profound proinflammatory phenotype, implying that Δ122p53 is a dominant oncogene[27]. Figure 1e illustrates immunohistochemical analysis of p53 in diffuse large B cell lymphomas (DLBCL) tissues from the Δ122p53 homozygous mice. Two major types of Δ122p53 localisation were detected: predominantly cytoplasmic aggregates (upper panel, Fig. 1e) or diffuse cytoplasmic (in some cases also nuclear) mixed with aggregates (lower panels, Fig. 1e).

The Δ122p53 aggregation observed in Δ122p53 mouse model were in agreement with those observed in human cells expressing Δ133p53β.

Next, we analysed the presence of Δ133p53β in human tumours. The Δ133p53β expression was first assessed in an array of breast cancers using RNAscope. As demonstrated in the upper panel of Fig. 1f metastatic breast cancer was found to be positive for the expression of the mRNA for this isoform. In addition, we analysed the presence of Δ133p53β protein in serial slices of the same tumour applying immunohistochemical approach using KJC8 antibody that specifically detects β isoforms[15]. In agreement with the above, Δ133p53β isoform also has a propensity to form protein aggregates in human breast cancers (Fig. 1f lower panel and digital magnification). Both H1299 cells expressing Δ133p53β and human metastatic breast cancer showed an identical pattern of expression for this isoform: a portion of diffuse staining was adjacent to pronounced aggregates. Even more interesting was the finding that these human tumour Δ133p53β aggregates were mostly located in the cytoplasm fully confirming observations obtained in the H1299 cell line.

In addition, we analysed the presence of the Δ133p53β protein aggregates in different human tumour types applying an identical immunohistochemical approach using KJC8 antibody. Δ133p53β expression was assessed in an array of primary breast, lung and colorectal cancers and their respective brain metastasis. Also, healthy associated breast and lung tissues were analysed. As shown in Fig. 1g and Supplementary Table 1 Δ133p53β aggregates were detected in most of the tumours analysed. Two major patterns of Δ133p53β aggregation were observed: some tumours had aggregates spread throughout the whole tumour while others only had aggregates in selected regions (the latter are highlighted with an asterisk in the Supplementary Table 1). Interestingly in breast tumour the presence of the aggregates was more obvious in the aggressive triple negative and Her2+ subtypes compared to the milder ER + Pr + HER2-negative Luminal A subtype (Supplementary Table 1 and Fig. 1g). In triple negative and Her2+ subtypes, aggregation tends to increase in the brain metastases compared to the corresponding primary tumours. In normal breast and lung associated tissue, aggregates were not detected except in some rare scattered normal cells in the breast samples probably reflecting lymphocytes. In addition,

brain metastasis of both lung and colorectal tumours showed aggregation while they were not observed in primary lung carcinoma (Supplementary Table 1 and Fig. 1g).

Our previous work had demonstrated the presence of the Δ133p53β aggregates in human prostate cancer[25]. To further evaluate this aggregation pattern in prostate cancer, RNAseq analysis conducted on 12 prostate cancers previously published[25] was used for gene set enrichment analysis. Next, we identified transcripts that are correlated with Δ133p53β with a Spearman's correlation cut-off of ≥0.3 or ≤−0.3. Using these transcripts, gene set enrichment analysis (FDR cut off <0.05) was carried out. Enrichment of Δ133p53β expression clearly correlated with cellular processes linked to homo-oligo and homo-tetramerisation confirming the presence of the aggregation phenomena in prostate cancer in the presence of the Δ133p53β isoform (Supplementary Fig. 2c, d). Examples of positive correlation with Δ133p53β include *RNF135* (Ubiquitin E3 ligase), *KCTD5* (Adapter for E3 ligase), *FUS* (RNA binding protein), *TRPM2* (involved in neurodegeneration via protein aggregation) and *CRTC2* (lysosomal pathway) (Supplementary Fig. 2e).

All together these data showed that Δ133p53β isoform aggregates in array of different primary human tumours and more markedly in their brain metastasis. It is interesting to observe that higher aggregation, which reflects increase expression of Δ133p53β isoform is associated with tumour aggressiveness.

Finally in order to test the aggregation properties of recombinant WT Δ133p53β we overexpressed it fused to Maltose Binding Protein (MBP-Δ133p53β) in *E. coli* (see "Methods" for details). After affinity purification, the size of MBP-Δ133p53β was estimated using size exclusion chromatography (SEC). The chromatogram showed a single peak that partially overlapped with the void volume of the column, and with an estimated molecular weight of 440 kDa, indicating that the purified protein corresponds to a large soluble species (Fig. 1h). SDS-PAGE clearly demonstrated that MBP-Δ133p53β is the only component of the fractions corresponding to the maximum of the peak and the void volume (Fig. 1h). The large size of these MBP-Δ133p53β oligomers was further validated by dynamic light scattering (DLS). The maximum of the size exclusion chromatogram presented a single DLS peak corresponding to particles with a broad range of diameters (from 10 to 40 nm) (Fig. 1i). MBP cleavage attempts were unsuccessful probably due to the lack of accessibility of the protease to the linker. In summary, the biophysical characterization of MBP-Δ133p53β indicates that this protein forms large oligomers of different sizes in solution.

In summary, these data reveal the aggregate forming capacity of WT Δ133p53β protein. Interestingly Δ133p53β aggregates appear to be different to those created by p53R175H since they were more denaturation resistant with an important non-soluble fraction. Immunofluorescence analysis revealed their presence both in cytoplasm and nucleus and partial co-localization with amyloid type aggregates but not intermediate oligomers. In addition, similar aggregates and cellular distribution were observed in the transgenic mouse model expressing Δ122p53 isoform. These observations were fully confirmed by staining of an array of human cancers demonstrating the presence of Δ133p53β aggregates in the human cancers. Finally, our in vitro analysis of recombinant protein confirmed the aggregate forming capacity of WT Δ133p53β isoform.

**The WT Δ133p53β has an unfolded conformation**. We next addressed how WT Δ133p53β protein has aggregation forming capacity. A large body of literature has demonstrated that unfolding of the p53R175H DNA-binding domain promotes aggregation-forming capacity. This unfolding leads to exposure of protein regions with amyloid-forming capacities, which are typically deeply buried within WT p53. This denaturation event could be detected by Ab240, a monoclonal α-p53 antibody to an epitope at position 213–217, when used in immunoprecipitation analysis. As shown in Fig. 2a Ab240 detected only p53R175H structurally mutated protein, but not native WT p53 or p53 proteins with contact mutations. The two latter proteins have an intact globular DNA-binding domain and the epitope is not accessible to Ab240. Surprisingly, WT Δ133p53β demonstrated identical immune-reactivity to Ab240 as a p53 structural mutant. More intriguingly, mutations (structural-R175H or contact-R280K) had no further effect on WT Δ133p53β immunoreactivity to Ab240 (Fig. 2b). In addition, Δ133p53α also exhibits immunoreactivity to Ab240 suggesting that this isoform has an unfolded DNA-binding domain similar to the p53R175H structural mutant and the Δ133p53β isoform. Also, we were able to demonstrate that WT Δ133p53α isoform is also immune-reactive to Ab240 (Supplementary Fig. 5d). Furthermore, we compared immune-reactivity to Ab240 of structural mutant p53R175H and WT Δ133p53β respectively to WT p53. As shown in Fig. 2c both mutant p53R175H and WT Δ133p53β proteins have significantly increased immune-reactivity to the antibody that detects unfolding events compared to the WT form of the p53 protein. These data suggest that WT Δ133p53β isoform has an unfolded DNA-binding domain similar to the p53R175H structural mutant.

In addition, the computational analysis suggested significant unfolding in WT Δ133p53β protein. The WT p53 consists of a well-structured central DNA-binding domain (DBD) (residues 94–294)[31] flanked by two intrinsically disordered regions. The N-terminal flanking region (residues 1–93) represents a transcription-activation domain, which adopts fixed conformations upon binding to transcription factors. The C-terminal tetramerization domains (residues 325–355) form the structure upon homo-oligomerization. In comparison with WT p53, the WT Δ133p53β isoform does not have an N-terminal transactivation region and the major part of the C-terminal tetramerization domain is truncated. Moreover, the WT Δ133p53β isoform does not have the first 39 residues of the central DBD (Fig. 2d). In silico analysis was performed to establish possible effects of this truncation on the structural state of the region 133–294 remaining from the DBD. The first consequence was the disappearance of several crucial contacts between the N- and C-termini of the DBD (Fig. 2e, f). For example, a salt bridge between K132 and E285, interactions between L130 and L289 and T125 and G279 are lost as indicated by circle in Fig. 2e. As a result, in the Δ133p53β isoform, the C-terminal α-helix, which is a key element of the DBD interface, may unfold and disrupt the binding with DNA. In addition, analysis of p53 protein using IUPRED program[32] shows that in the full-length protein the C-terminal α-helix represents the region with the highest tendency to be unfolded (Fig. 2d). Furthermore, the absence of the N-terminal region 94–132 in the Δ133p53β isoform may also destabilise the β-sandwich core of the DBD. Indeed, the IUPRED predicts that this region has the highest tendency to be folded (Fig. 2d). In the p53 DBD structure, this N-terminal fragment caps the hydrophobic core of the DBD β-sandwich structure. Its removal will expose the hydrophobic residues M133, F134, V135, V141, V143, V145, I255, L257, L265, F270 (Fig. 2e–g) of the β-sandwich core to the solvent, leading to destabilization of the DBD.

These analyses suggest that the remaining part of the DBD either has a very low structural stability or, most probably is unfolded. If WT Δ133p53β isoform is unfolded, this increases its aggregation potential, because in the majority of cases a

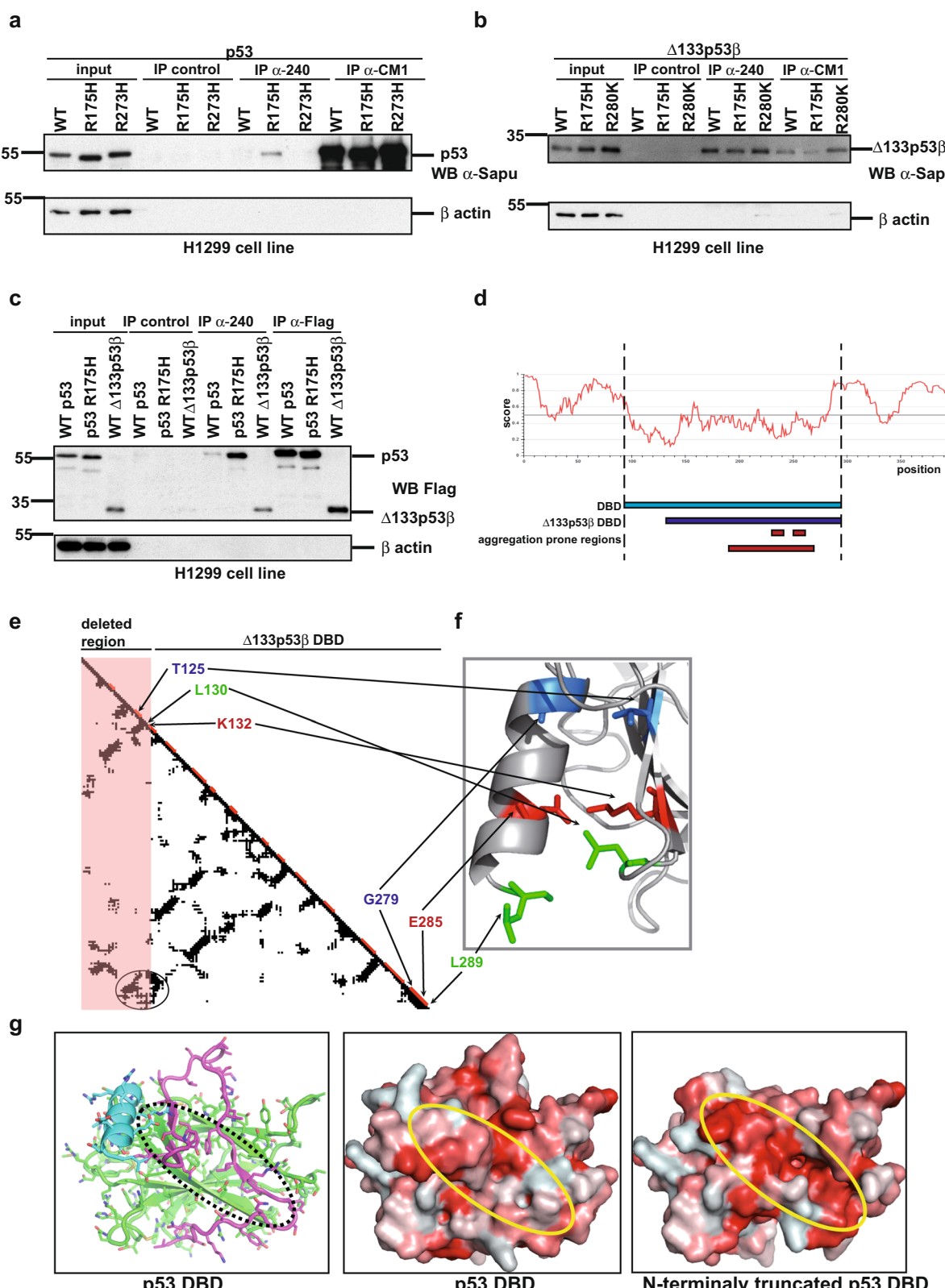

polypeptide chain with aggregation-prone motifs must be unfolded prior to the aggregation[33]. Our analysis by using several programs for prediction of aggregation-prone regions shows that the WT Δ133p53β sequence has a number of such domains (Fig. 2d).

Taken together, these data clearly show that WT Δ133p53β has an unfolded conformation similar to p53R175H structural mutant.

**CCT chaperone complex regulates aggregation-forming ability of WT Δ133p53β**. To evaluate which proteins could be involved in folding and aggregation of the Δ133p53β isoform, we adopted a high-throughput proteomic approach. Δ133p53β was transfected into H1299 cell line and upon co-immunoprecipitation and proteomic analysis an array of partners was detected. As shown in Supplementary Fig. 5e we were able to detect immunoprecipitated

**Fig. 2 Characterisation of WT Δ133p53β isoform secondary structure. a** Immunoprecipitation analysis of p53 WT, structurally and contact mutated proteins by Ab 240 and CM1 in H1299 cell line. $n = 3$. **b** Immunoprecipitation analysis of Δ133p53β WT, structurally and contact mutated proteins by Ab 240 and CM1 in H1299 cell line. $n = 3$. **c** Immunoprecipitation analysis of WT and structural mutant of p53 and WT Δ133p53β by Ab 240 and CM1 in H1299 cell line. $n = 3$. **d** IUPRED program analysis for prediction of the tendency to be unstructured performed on Δ133p53β protein. Amyloidogenic regions are predicted by using Waltz[67]—region 232–237 (IHYNYM); PASTA[68] and ZipperDB[50]—region 251–257 (ILTIITL) and ArchCandy with threshold 0.370[33]—region 195–270. **e** Contact map of WT p53 DBD was realized with CMview software[69] using the structure with PDB ID 1TUP. Horizontal and vertical axes of the plot correspond to the amino acid sequence of protein and a point located at the intersection of two different residues indicates a putative contact with the distance between two atoms of these amino acids ≤8 Å. Elements of the secondary structures are shown in red on the diagonal. The first 39 residues depleted in Δ133p53β are outlined by a pink rectangle area. The C-terminal α-helix is stabilised by the cluster of contacts encircled in the left-down corner of the map. Most of these contacts come from residues located in the depleted fragment. Among these contacts we distinguish ones that should play a key role in the α-helix stabilisation: T125-G279 (in red), L130-L289 (in green), K132-E285 (in blue). **f** Zoom on the contacts between the N-terminal region and C-terminal α-helix. Side chains involved in the important contacts are coloured. **g** Left panel: the 3D structure of p53 DBD with the N-terminal region 94–132 in magenta and the C-terminal α-helix (276-289) in blue. Middle panel: Surface of the p53 DBD. Right panel: Hypothetical surface of the DNA-binding domain of Δ133p53β isoform if it would be structured. The surfaces are colour-coded by hydrophobicity of amino-acid residues using scale from ref. [70]. For the sake of comparison, all three structures have the same orientation. Oval regions over the structures denote area of the hydrophobic core of the DBD β-sandwich fold. The truncated structure of p53 DBD was generated and analysed by using PyMol program[71]. Molecular weight is expressed in kDa. Source data are provided as a Source Data file.

partners already on silver-stained polyacrylamide gels so we focused our proteomic analysis on this region as depicted in Supplementary Fig. 5f. As chaperones are known to be common components of cellular aggregates across various contexts[34], we focused on chaperone partners of Δ133p53β. Interestingly the CCT complex was found as a partner of Δ133p53β isoform. This complex is involved in refolding of newly synthesized proteins but also in the prevention of aggregation. It consists of two stacked rings of eight paralogous subunits (CCT1 to CCT8) each[35]. Interaction of the CCT complex with Δ133p53β isoform was confirmed by co-immunoprecipitation analysis (Fig. 3a). To determine which regions of Δ133p53β are involved in CCT subunits interaction, co-immunoprecipitation analysis was done after transfection of Δ133p53β, Δ133p53α or Δ160p53β isoforms in H1299 cells. Δ133p53β and Δ133p53α differ in their C-terminal ends, whereas Δ133p53β and Δ160p53β differ in the N-terminus. Δ160p53β could be considered to be a deleted form of Δ133p53β (Supplementary Fig. 5c). Δ133p53α associates with the CCT3 subunit, but with lower affinity compared to Δ133p53β. This indicates modest involvement of the C-terminus for subunit 3 of CCT complex (CCT3) interaction. However, the loss of the N-terminal 27 amino acids from Δ133p53β almost completely abolished binding to CCT3, as Δ160p53β was found to be poorly bound to this subunit (Fig. 3b). In addition, we compared affinities of WT p53 as well as its structural R175H mutant and Δ133p53 isoforms to the CCT complex. As shown in Fig. 3b Δ133p53 isoforms and particularly Δ133p53β have drastically increased affinity to the CCT complex respect to the TA isoforms.

These data indicate that the N-terminus of Δ133p53 isoforms is critical for binding the CCT complex.

To further investigate the involvement of the CCT complex in Δ133p53β aggregation, we depleted three subunits of the complex, CCT3, CCT5 and CCT7 (Fig. 3c). Depletion of the CCT complex resulted in a marked increase of Δ133p53β aggregation both in the soluble and most markedly, in the insoluble protein fraction (Fig. 3d, e, respectively). Accordingly, immunofluorescence analysis exhibited important accumulation of Δ133p53β aggregates upon CCT complex depletion (Fig. 3f). Similarly, aggregate accumulation was reported for full-length WT p53 after CCT chaperone depletion[11].

To evaluate if the aggregation status of Δ133p53β impacts its function, Δ133p53β isoform was expressed both in the presence or absence of the CCT complex and the effect on migration of H1299 cells was evaluated. Whereas Δ133p53β increased migration of H1299 cells, depletion of CCT complex by shRNA treatment for CCT2 and CCT3 subunits, completely abolished the

migration effect of Δ133p53β. Depletion of the CCT complex in the absence of Δ133p53β isoform had no effect on the basal level of H1299 cell migration capacity (Fig. 3g–i).

To investigate whether regulation of Δ133p53β aggregates is specific to CCT complex or depends on interactions with other chaperones we tested effects of the association of HSP70 and Pontin on Δ133p53β aggregation state. Interestingly, both proteins associate only with mutated but not WT p53[36–39].

Using a co-immunoprecipitation approach we were able to confirm that Pontin interacts only with mutated p53 (Supplementary Fig. 3a) while HSP70 associates only with structurally mutated p53R175H (Supplementary Fig. 3b). Similarly, WT Δ133p53β, contact (Δ133p53βR280K) or structural (Δ133p53βR175H) mutants were transfected into H1299 cells and immunoprecipitation analysis carried out. All Δ133p53β proteins bounds to Pontin and HSP70, independently of their mutation status (Supplementary Fig. 3c, d) demonstrating features similar to mutated p53. Next, we compared the affinities of Δ133p53β and a structural p53 mutant to Pontin. In order to do this WT or Δ133p53β with contact mutation (R280K) were transfected into H1299 cells as well as the p53R175H structural mutant and immunoprecipitation analysis was performed. As shown in Supplementary Fig. 3e all transfected proteins demonstrated an identical affinity to Pontin.

To evaluate if this association occurs in 'natural' cellular backgrounds, each of Δ133p53β isoforms was expressed in the cell of origin and identical analysis was performed. Namely, WT Δ133p53β was transfected in MCF-7, Δ133p53βR280K in MDA-MB-231 D3HLN and Δ133p53βR175H in SK-BR-3 cell line. All Δ133p53β proteins interact with Pontin and HSP70 in their natural cellular backgrounds, independently of their mutational status (Supplementary Fig. 3f, g for Pontin, Supplementary Fig. 3h–j for HSP70).

Finally, we evaluated if Pontin and HSP70 are involved in the aggregation of Δ133p53β isoform. In order to test this, we depleted those proteins by using Sh RNAs and tested the aggregation of Δ133p53β in this cellular background. As shown in Supplementary Figs. [Fig. 3k–m (for Pontin) and 3n–p (for HSP70)] depletion of these proteins had no important effect on Δ133p53β aggregation capacity.

All together these data demonstrated the critical involvement of the CCT complex in Δ133p53β aggregation capacity and consequently on Δ133p53β-promoted cell migration. On the other hand, although shown to interact with Δ133p53β isoform, Pontin and HSP70 are not involved in Δ133p53β aggregation.

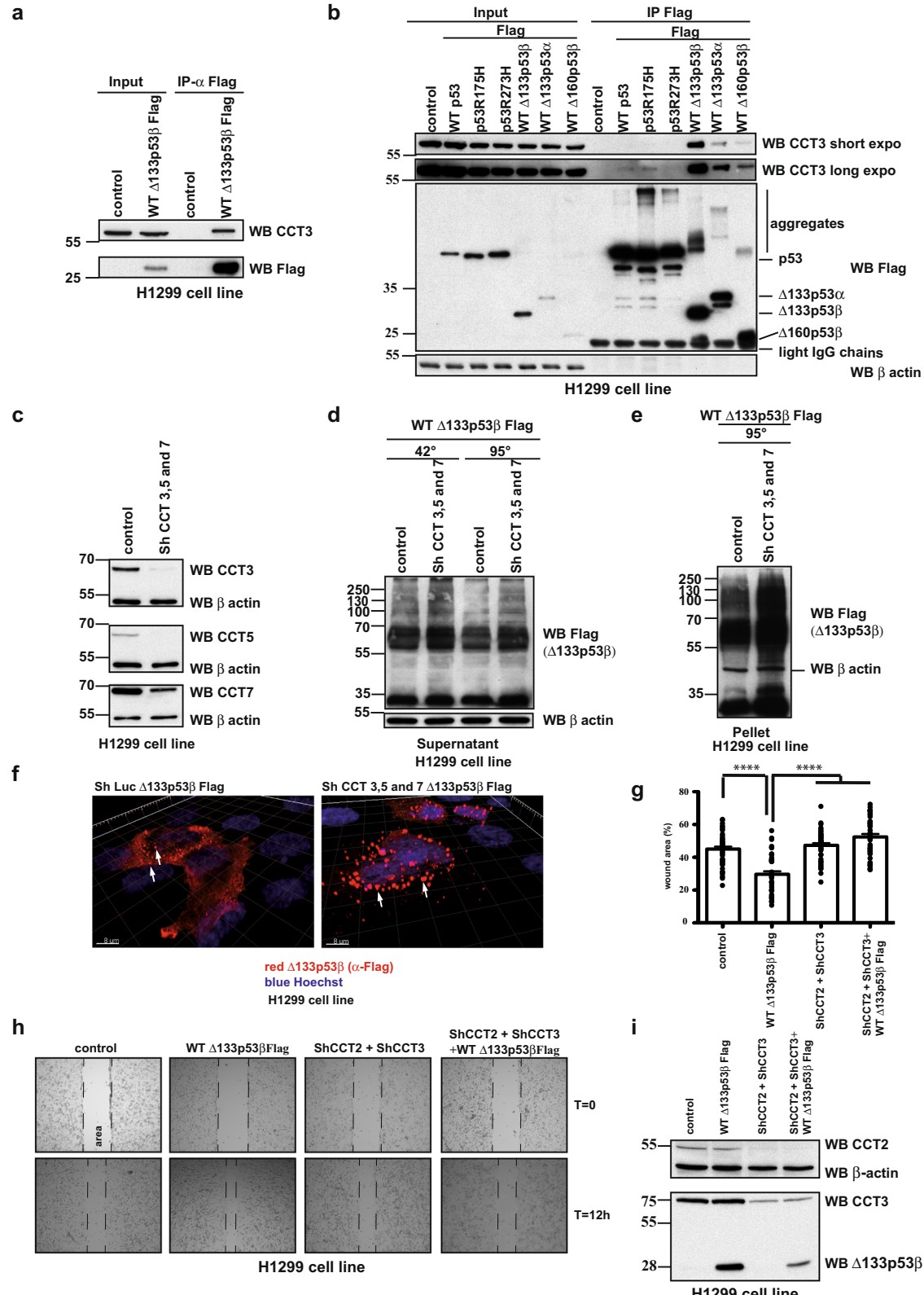

**Interaction with p63 family members reduces aggregation of Δ133p53β**. One of the features of mutant p53 aggregates is co-aggregation with interacting partners, which heavily influences cancer cell biology. This is notably the case of co-aggregation of p53 with family members p63 and p73[10]. To investigate whether the Δ133p53β can co-aggregate with p53, we first evaluated if it interacts with this protein. To do this immunoprecipitation

analysis was carried out. While Δ133p53α either in WT or mutated form in different breast cancer cell lines associated with WT or mutated p53, the interaction between Δ133p53β and p53 was not detected (Fig. 4a–c). Furthermore, we extended this analysis to colon cancer cell lines HCT116 (WT p53) and SW480 (R273H/P309S mutant). In HCT116 and SW480 cell lines Δ133p53β has identical interacting partners to breast cancer cell

**Fig. 3 CCT complex actively participate in regulation of Δ133p53β aggregation and its activity. a** Co-immunoprecipitation analysis of CCT3 subunit and WT Δ133p53β in H1299 cells. *n* = 3. **b** Co-immunoprecipitation analysis of CCT3 subunit with different p53 and Δ133p53 isoforms in H1299 cells. *n* = 3. **c** Western blot analysis of CCT complex subunit depletion in H1299 cells. *n* = 2. **d** Western blot analysis of the soluble protein fraction from H1299 cells after WT Δ133p53β transfection in control and CCT3, 5 and 7 depleted backgrounds. *n* = 3. **e** Western blot analysis of the insoluble protein fraction from H1299 cells after WT Δ133p53β transfection in control and CCT3, 5 and 7 subunit depleted backgrounds. *n* = 3. **f.** Immunofluorescent analysis of WT Δ133p53β aggregates in control and CCT3, 5 and 7 subunits depleted H1299 cell line upon 3D reconstruction of the confocal Z-stacks. Red: α-Flag (Δ133p53β), blue: Hoechst. Scale bar 8 μm. *n* = 3. **g** Quantification of the migration ability of the H1299 cell expressing Control, WT Δ133p53β or CCT complex depleted background at *t* = 12 h after wound-healing, *n* = 3. Values are the mean of ±SEM (error bars) pooled from three independent experiments with 15 images for each sample in each experiment (total 45 images/sample). Statistical analysis was performed using non-parametric Mann–Whitney *t* test with the Prism software (GraphPad). All *p* values are <0.0001. **h** Phase contrast images of wound-healing migratory assay of H1299 cells expressing WT Δ133p53β in control or CCT complex depleted background at the beginning (*T* = 0 h) and end of the test (12 h). Representative images of one out of three experiments are shown. **i** Western blot analysis of CCT complex subunit depletion in H1299 cells. Representative immunoblot of one out of three experiments from Fig. 3h is shown. The arrows indicate aggregates. Molecular weight is expressed in kDa. Source data are provided as a Source Data file.

lines, notably the CCT complex and HSP70, while again an interaction with p53 was not detected (Supplementary Fig. 5a).

Since Δ133p53β does not associate with p53 we next evaluated its interaction with paralogs from the p63 family. p63 protein isoforms, encoded by the *TP63* member of the p53 gene family, also fall into two major subclasses, TAp63 (α to ε) isoforms, which harbour the N-terminal transactivation domain, and ΔNp63 (α to ε) isoforms, which are depleted for a part of the N-terminal transactivation domain. It is claimed that TAp63 isoforms have tumour- and metastasis-suppressor activities, whereas ΔNp63 isoforms promote cancer progression[40]. The current view is that ΔNp63 isoforms act as dominant-negative variants that counteract the transcriptional activity of TAp63 isoforms[40,41]. When co-expressed in H1299 cells, WT Δ133p53β and ΔNp63α bind to each other (Fig. 4d). Interestingly CCT3 protein also associates with this complex (Fig. 4d). Next, we evaluated the aggregation status of the WT Δ133p53β upon co-expression of ΔNp63α by a modified western blot assay and immunofluorescence analysis. Co-expression of ΔNp63α significantly reduced aggregation of WT Δ133p53β both in the soluble (Fig. 4e) and insoluble protein fractions (Fig. 4f). In addition, immunofluorescence analysis upon confocal microscopy and 3D reconstruction of z-stacks, shows the absence of aggregates when WT Δ133p53β was co-expressed with ΔNp63α. Furthermore, this analysis demonstrated co-localisation of both proteins in the nucleus (Fig. 4g). Finally, we performed time-lapse microscopy on the cells co-expressing WT Δ133p53β and ΔNp63α (Supplementary movie 3) or expressing either WT Δ133p53β (Supplementary movie 1) or ΔNp63α (Supplementary movie 2) alone. The presence of ΔNp63α (fused with mCherry) reduced the aggregate forming capacity of WT Δ133p53β. Of note, no aggregation of the ΔNp63α or TAp63α was observed during this analysis even in conditions of overexpression (Fig. 4e and g, i panels "WB p63" and "ΔNp63α", respectively and Supplementary movies 2 and 3), suggesting that this phenomenon is a hallmark of WT Δ133p53β alone.

We next examined if other members of the p63 family produce similar effects on the aggregation capacity of WT Δ133p53β. To do this we tested if TAp63α protein interacts with WT Δ133p53β. As shown in Fig. 4h upon co-expression and co-immunoprecipitation analysis we detected interaction of these two proteins. Again, association of CCT3 protein was detected both with WT Δ133p53β alone and in presence of TAp63α. Similarly to ΔNp63α, TAp63α affects the aggregation state of WT Δ133p53β. Co-expression of TAp63α completely abolished WT Δ133p53β aggregates in both the soluble and insoluble protein fractions (Fig. 4i, j).

**Δ133p53β confers invasiveness and EMT features through p63 isoforms.** These data confirm that members of the p63 family

interact with WT Δ133p53β and have a strong effect on its aggregation ability. Consequently, we tested if these interactions impact cancer cell motility. We first evaluated the expression of the p63 isoforms in different cancer cell lines by RT-PCR. p63 isoforms were detected using primers listed in Supplementary Table 6 and nested PCR was performed according to the combination of the primers indicated in Supplementary Table 7. This analysis demonstrated that only MCF-7 cells express all p63 family members, except TAp63ε (Fig. 5a). Next, we evaluated if WT Δ133p53β interacts with some of these proteins in MCF-7 cells. Co-expression analysis demonstrated that WT Δ133p53β associates with ΔNp63α and ΔNp63γ isoforms in this cell line (Fig. 5b). Finally, we evaluated if this interaction affects the invasive capacities of MCF-7 cells. The expression of WT Δ133p53β alone resulted in increased invasive capacity in MCF-7 cells, which is in agreement with previously published data[22]. Furthermore, co-transfection of WT Δ133p53β with either ΔNp63α or ΔNp63γ, revealed a highly synergistic effect (Fig. 5c). Since overexpression of WT Δ133p53β induces effects on MCF-7 cells that are phenotypically similar to the epithelial to mesenchymal transition (EMT) (see Supplementary movies 4–9) we evaluated hallmarks of this phenomenon upon co-expression of WT Δ133p53β and ΔNp63 isoforms. As shown in Fig. 5d, expression of WT Δ133p53β alone had a small effect on E-cadherin expression whereas expression of ΔNp63α and ΔNp63γ alone had no effect. However, co-expression of WT Δ133p53β and ΔNp63α strongly reduced E-cadherin and increased N-cadherin level, which is in complete agreement with the synergistic effect of these two isoforms. Interestingly, depletion of endogenous ΔNp63 isoforms significantly reduced invasiveness conferred by ectopic Δ133p53β expression in MCF-7 cells, suggesting that WT Δ133p53β-mediated invasiveness is dependent on endogenous ΔNp63 in MCF-7 cells (Fig. 5e, f). Collectively, these data indicate that WT Δ133p53β forms a protein complex with ΔNp63α or ΔNp63γ and acts in synergy with them to drive EMT, as characterised by the loss of epithelial features and gain of invasive properties in MCF-7 cells.

Interestingly TAp63 depletion, either alone or concomitantly with depletion of Δ133p53 or β isoforms did not change invasiveness, indicating that TAp63 expression does not alter the invasive activity of MCF-7 cells (Fig. 5g). The lack of effect of the TAp63 isoforms or Δ133p53 depletion may be due to the very low invasive capacity of MCF-7 and also to the low basal level of Δ133p53β in MCF-7 (Supplementary Fig. 4a) We then investigated whether TAp63 and Δ133p53β are involved in the invasiveness of highly invasive MDA-MB-231 D3H2LN cells. The MDA-MB-231 D3H2LN cells endogenously express TAp63α and TAp63β, but not the ΔNp63 isoforms (Fig. 5a). In addition, MDA-MB-231 D3H2LN cells were previously shown to be significantly more invasive than MCF-7 cells[22] and they express

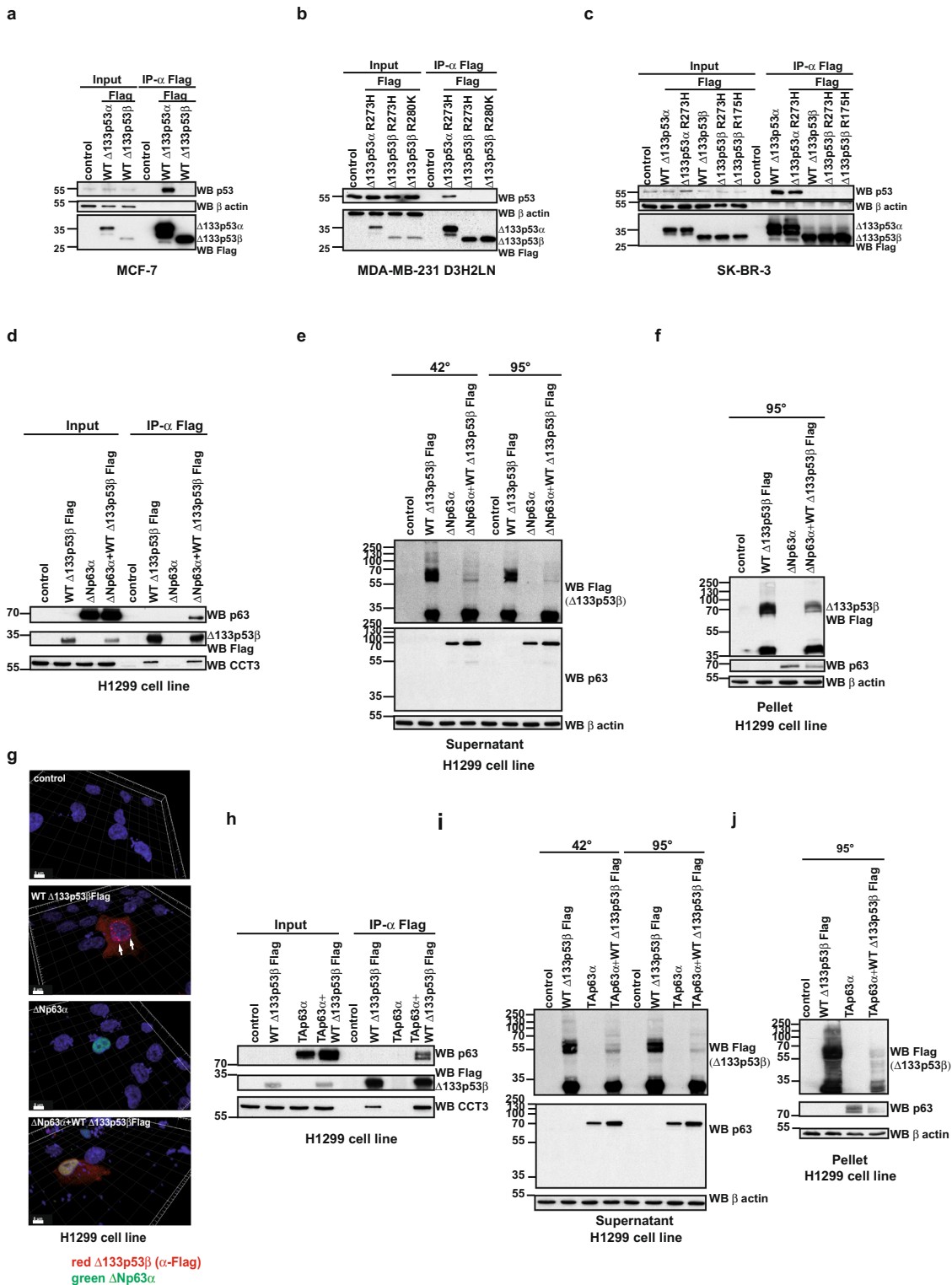

increased levels of Δ133p53β isoforms compared to MCF-7 (Supplementary Fig. 4a). Of note levels of CCT complex subunits are also increased in highly invasive MDA-MB-231 D3HL2N compared to weakly invasive MCF-7 cells (Supplementary Fig. 4b).

Ectopic expression of TAp63α decreased cell invasion while depletion of endogenous TAp63 isoforms by shRNA significantly enhanced invasiveness of MDA-MB-231-D3H2LN cells, indicating that invasiveness of these cells is dependent on endogenous TAp63

(Fig. 5h, i). As expected, depletion of endogenous Δ133p53 isoforms reduced invasiveness (Fig. 5i). However, the reduction of invasiveness conferred by depletion of endogenous Δ133p53 isoforms was no longer impacted by depletion of endogenous TAp63 isoforms suggesting that these two proteins might operate in the same signalling pathway. However, depletion of endogenous Δ133p53 isoforms drastically reduced the increased invasiveness conferred by depletion of endogenous TAp63 isoforms alone (Fig. 5i). These data show that Δ133p53-driven invasiveness depends on endogenous

**Fig. 4 The loss of the aggregation of the WT Δ133p53β upon interaction with p63 family members. a** Co-immunoprecipitation analysis of WT Δ133p53α and WT Δ133p53β interaction with endogenous p53 upon expression in MCF-7 cells. $n = 5$. **b** Co-immunoprecipitation analysis of mutated Δ133p53α and Δ133p53β interaction with endogenous p53 upon expression in MDA-MB-231 D3H2LN cells. $n = 5$. **c** Co-immunoprecipitation analysis of mutated Δ133p53α and Δ133p53β interaction with endogenous p53 upon expression in SK-BR-3 cells. $n = 5$. **d** Co-immunoprecipitation analysis of WT Δ133p53β and ΔNp63α interaction upon co-expression in H1299 cells. $n = 3$. **e** Western blot analysis of the aggregate forming capacity of WT Δ133p53β in the soluble protein fraction from H1299 cells upon co-expression with ΔNp63α protein. $n = 3$. **f** Western blot analysis of the aggregate forming capacity of WT Δ133p53β in the insoluble protein fraction from H1299 cells upon co-expression with ΔNp63α protein. $n = 3$. **g** Immunofluorescent analysis of WT Δ133p53β aggregates after co-expression with ΔNp63α in H1299 cells upon 3D reconstruction of the confocal Z-stacks. Red: α-Flag (Δ133p53β), green: ΔNp63α, blue: Hoechst. Scale bar 8 μm. $n = 3$. **h** Co-immunoprecipitation analysis of WT Δ133p53β and TAp63α interaction upon co-expression in H1299 cells. $n = 3$. **i** Western blot analysis of the aggregate forming capacity of WT Δ133p53β in the soluble protein fraction from H1299 cells upon co-expression with TAp63α protein. $n = 3$. **j** Western blot analysis of the aggregate forming capacity of WT Δ133p53β in the insoluble protein fraction from H1299 cell line upon co-expression with TAp63α protein. $n = 3$. The arrows indicate aggregates. Molecular weight is expressed in kDa. Source data are provided as a Source Data file.

TAp63 protein isoforms in triple-negative MDA-MB-231 D3H2LN cells. Furthermore our data support observations from the literature that TAp63 and ΔNp63 act in opposition in cancer cell invasion[40].

Altogether these data demonstrated that loss of Δ133p53β aggregation upon interaction with p63 family members significantly affects key functions of the cancer cell like invasive capacities. These experiments strongly suggest that the activity of Δ133p53β depends on its aggregation state, which is modulated by interactions with protein partners.

**Interaction with p63 family members increases intracellular Δ133p53β kinetics.** To support the notion that the dynamics of Δ133p53β aggregation depend on the presence of their partners we applied the Fluorescence Recovery After Photobleaching (FRAP) technique. This approach evaluates the kinetics of the molecules re-filling the photobleached region of the cell. To this end, Δ133p53βEGFP and mCherryΔNp63α were expressed in H1299 cells. Recovery of fluorescence upon photobleaching was measured in samples expressing EGFP (control), Δ133p53βEGFP (condition of aggregates) or co-expression of Δ133p53βEGFP and mCherryΔNp63α (condition of aggregate dissolution). A flow chart of the FRAP experiment is depicted in Fig. 6a. As shown in Fig. 6b, photobleaching demonstrated highly increased recovery of Δ133p53βEGFP in the presence of mCherryΔNp63α compared to Δ133p53βEGFP alone. Intracellular mobility of Δ133p53βEGFP is significantly increased when mCherryΔNp63α is co-expressed and reached values close to those obtained with the diffuse protein GFP used as control (Fig. 6b). Furthermore, linear regression factor with $R^2$ with no less of 0.95 were compared demonstrating significant difference between Δ133p53βEGFP alone or in presence of the mCherryΔNp63α partner (Fig. 6c). Immunofluorescence analysis (Fig. 6d) showed that Δ133p53β and ΔNp63α fused with fluorescent tags had identical co-localisation as without tags (Fig. 4g).

To determine the recovery of fluorescence in the bleached area over time, images were obtained before and after photobleaching. Figure 6e shows time-lapse recovery 30 s upon nuclear photobleaching (from Supplementary movies 10–12). FRAP analysis of Δ133p53βEGFP thus revealed that the fluorescence within bleached aggregates recovered slowly.

Altogether these results fully confirmed previous observations that partners of Δ133p53β isoform, like ΔNp63α, are recruiting this isoform from aggregates, increasing its intracellular mobility and thus making it available for different cellular functions like migration and invasion.

**Discussion**

*TP53* encodes at least 12 different protein isoforms. Recent studies have suggested that Δ133p53β contributes to several aspects of cancer progression, such as invasion, migration and the cancer stem cell phenotype[22,26]. Consistent with this, various studies have reported high level of Δ133p53β correlated with decreased survival and worse prognosis in different human cancers[20,22–25]. However, the mechanisms by which Δ133p53β carries out its oncogenic functions are still largely unknown. Here we identify a p53-independent pro-invasive mechanism of Δ133p53β that relies on the aggregation state of this protein and interactions with specific partners like CCT complexes and ΔNp63. We show Δ133p53β protein aggregates occur in several human cancer cell lines. This observation was fully confirmed by the detection of aggregates in a wide array of human tumour biopsies and in lymphoma tissue from the *Δ122TP53* mice. In addition, we demonstrated that in vitro recombinant WT Δ133p53β protein forms large oligomers of different sizes in solution. Experimental and computational approaches strongly suggest that the aggregate forming capacity of Δ133p53β is associated with the unfolding of its DBD, most likely due to a very low structural stability of the remaining part of this region. The depletion of the N-terminal part and particularly 39 residues of DBD critically affected the double beta sheet conformation of the DBD and the α-helices that are responsible for contact with DNA. The computational analysis detected several possible domains in the WT Δ133p53β protein with the capacity to create amyloid-type aggregates. This is in agreement with a recent report suggesting that aggregation of the Δ40p53 isoform results from loss of its p53 N-terminal transactivation domain[42].

The unfolding of the protein structure is generally a prerequisite for aggregate formation. Using a high-throughput proteomic approach, we identified an association of the CCT chaperone complex with WT Δ133p53 isoforms. Importantly our data indicate that the N-terminus of Δ133p53 isoforms is crucial for binding the CCT complex. This chaperone was demonstrated to be involved in correct p53 folding[11]. Similar to p53, depletion of the CCT complex led to the accumulation of WT Δ133p53β aggregates in cells. Strikingly, depletion of the CCT complex also caused a loss of the WT Δ133p53β induced cell migration. Together these observations indicate that the CCT complex has an active role in WT Δ133p53β activation, while its depletion leads to aggregate accumulation and loss of Δ133p53β induced cell migration. Interestingly, we also identified Pontin and HSP70 as partners of Δ133p53β. Both proteins are thought to have chaperone activities[43–45], but they had no effect on WT Δ133p53β aggregation capacity.

A growing number of diseases are associated with incorrect protein accumulation and aggregation[46]. This mechanism is largely been described for neurodegenerative disorders like Alzheimer, Parkinson's or Huntington disease, but it is emerging as a crucial mechanism in oncogenesis[47]. Recent studies demonstrate the ability of mutant p53 to create aggregates[48,49]. Interestingly not all types of mutations have identical effects on p53 aggregation. So-called "contact" mutations affecting residues responsible

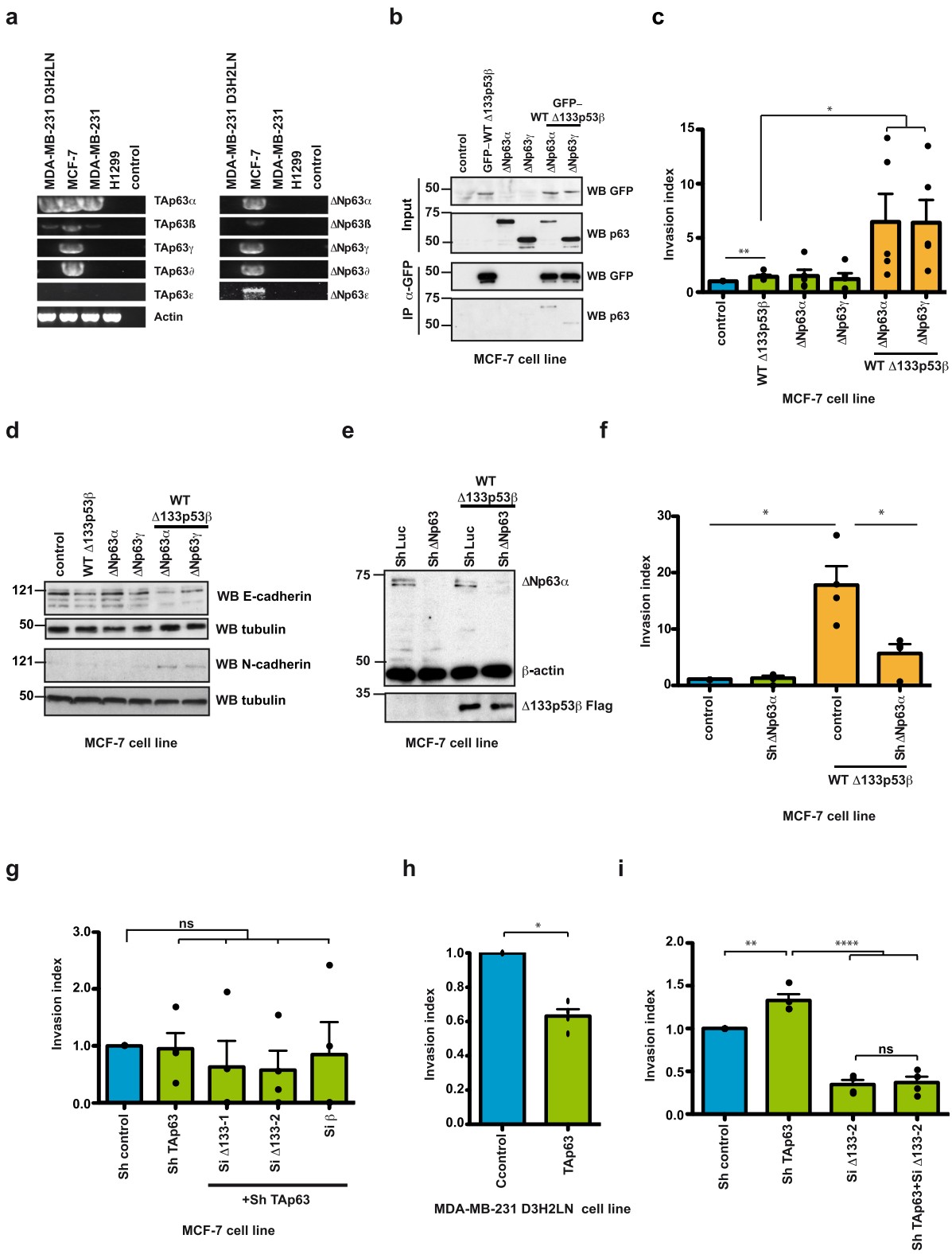

for DNA interaction have generally mild effects on aggregation propensity. On the contrary structural mutation affecting globular folding of the DNA-binding domain have drastic effects on aggregation[10]. One possible explanation for this is that unfolding of the DNA-binding domain leads to exposure of novel domains that are buried in the WT structure. These de novo exposed domains are typically responsible for the gain of function of

structurally mutated p53 proteins and aggregation may contribute to this phenomenon. The epitope at position 252–258 has been suggested to be particularly aggregation prone[50]. Correlation between mutant p53 aggregation, functional loss, and tumour growth has been also shown[51–53]. Interestingly mutant p53 aggregation was demonstrated to be a potential therapeutic target for cancer treatment. A recent study by Soragni and colleagues

**Fig. 5 WT Δ133p53β interaction with p63 family members modifies cancer cell invasive capacities. a** RT-PCR analysis of p63 isoform expression in different cancer cell lines. $n = 3$. **b** Co-immunoprecipitation analysis of GFP- WT Δ133p53β interaction with ΔNp63α or ΔNp63γ in MCF-7 cells upon co-expression. $n = 2$. **c** Invasion of MCF-7 breast cancer cells expressing Δ133p53β, ΔNp63α or ΔNp63γ either alone or in combination as indicated. Invasion assays were performed for 48 h upon protein expression. The results are expressed as the fold change compared to control. Values are the mean of ±SEM (error bars) of five independent experiments. Statistical analysis was performed using non-parametric Mann–Whitney $t$ test with the Prism software (GraphPad). p values are 0,0075, 0,0317 and 0,0159 respectively in the order of the appearance in the figure. **d**. Western blot analysis of E- and N-Cadherin in MCF-7 cells expressing GFP-tagged Δ133p53β, ΔNp63α or ΔNp63γ, either alone or in combination as indicated. $n = 2$. **e** Western blot analysis of ΔNp63 depletion upon Sh RNA application in MCF-7 cell line. Representative immunoblot of one out of 3 experiments from Fig. 5f is shown. **f** Invasion of MCF-7 breast cancer cells expressing with Δ133p53β in control and ΔNp63 depleted background as indicated. MCF-7 cells were assayed for 48 h and the changes in the invasion were analysed as in (**c**). The results are expressed as the fold change ratio compared to control. The values plotted are means ± SEM of four independent experiments. Statistical analysis was performed using non-parametric Mann–Whitney $t$ test with the Prism software (GraphPad). p values are 0.0211 and 0.286. **g** Invasion of MCF-7 breast cancer cells in TAp63 or double depleted (TAp63 and Δ133p53 and TAp63 and β) background as indicated. MCF-7 cells were assayed for 48 h and the changes in invasion were analysed as in (**c**). The results are expressed as the fold change ratio compared to control. The values plotted are means ± SEM of four independent experiments. Statistical analysis was performed using non-parametric Mann–Whitney $t$ test with the Prism software (GraphPad). **h** Invasion of MDA-MB-231 D3H2LN breast cancer cells under conditions of overexpression of TAp63α. Cells were assayed for 48 h and the changes in invasion were analysed as in (c). The results are expressed as the fold change ratio compared to control. The values plotted are means ± SEM of four independent experiments. Statistical analysis was performed using non-parametric Mann–Whitney $t$ test with the Prism software (GraphPad). p value is 0.0211. **i** Invasion of MDA-MB-231 D3H2LN breast cancer cells in TAp63α and Δ133p53 depleted background as well as in double depleted background. Cells were assayed for 48 h and the changes in invasion were analysed as in (**c**). The results are expressed as the fold change ratio compared to control. The values plotted are means ± SEM of four independent experiments. Statistical analysis was performed using non-parametric Mann–Whitney $t$ test with the Prism software (GraphPad). The first p value is 0,0061 while the other two are <0.0001. Molecular weight is expressed in kDa. Source data are provided as a Source Data file.

demonstrated that inhibition of the mutant p53 aggregation significantly reduced tumour progression[50]. Mutant p53 that has gain of oncogenic functions was demonstrated to result from co-aggregation with different tumour suppressors including p53 itself and its paralogs p63 and p73[10]. Interestingly WT p53 does not interact with p63 and its isoforms while p53R175H demonstrates strong affinity for the above-mentioned proteins[54].

We also noticed that the WT Δ133p53 isoforms recapitulate similar pro-oncogenic functions of mutant p53, and particularly the structurally mutated p53 in cancer progression. Namely WT Δ133p53β and structurally mutated p53 possess aggregate forming capacity due to the unfolded protein conformation. In addition, these proteins share identical cellular localisation, are both present in cytoplasm and nucleus, contrary to WT p53, which is detected only in the nucleus. Our study shows that WT Δ133p53β and structurally mutated p53 also share specific protein partners (Pontin, HSP70 and p63 family members), which do not bind to WT p53, as already described[36,39,54]. Interestingly, this isoform also had gain of functions typical for p53R175H such as stimulation of cancer cell migration. The unfolding event provides WT Δ133p53β with features similar to those described as gain of function for p53R175H mutant. Typically, newly exposed domains are available for association with the protein partners that do not bind WT p53, which leads to activation of new transcription or signalling pathways. This suggests that WT Δ133p53β and structurally mutated p53 share identical functions and regulations.

However, other features indicate that mutant p53 and WT Δ133p53β aggregates are structurally and functionally distinct. First, it has been largely accepted that the dominant-negative inactivation of WT p53 results from co-aggregation with mutant p53, which inhibits WT p53 activity and explains the dominant-negative effect of mutant p53[10]. As our data indicate that Δ133p53β cannot interact with WT p53, abrogation of WT p53 tumour suppressor functions in Δ133p53β dependent tumours cannot be explained by the same co-aggregation mechanism. Another plausible mechanism to explain the loss of wild-type p53 functions in Δ133p53β dependent tumours is that Δ133p53β may compete with WT p53 to bind p53 response elements on DNA. This hypothesis is strengthened by recent observation showing that Δ133p53β could bind p53 specific DNA[25,55].

Association of the WT Δ133p53β with several p63 isoforms in different cellular backgrounds was found to affect its aggregation status. In contrast to the reported co-aggregation of p63 and p53R175H, association of the p63 isoforms with WT Δ133p53β drastically reduced its aggregation capacity. Surprisingly, association of these proteins exerts strong effects on the invasive capacities of breast cancer cell line MCF-7 and MDA-MB-231 D3H2LN. Association of WT Δ133p53β and ΔNp63α resulted in the loss of the aggregates, which was followed by strong induction of cancer cell invasion. These effects were followed by phenotypic changes to the MCF-7 cell line similar epithelial-mesenchymal transition (EMT). While individual expression of the two proteins had no effect on E and N cadherin levels, their co-expression led to the loss of the former and induction of the latter. The cadherin switch is a classical feature of EMT. Together these experiments demonstrate both activation of the Δ133p53β invasive activity and aggregate loss upon co-expression with ΔNp63α isoform. These results are in agreement with previous reports demonstrating the interplay between Δ133p53 and ΔNp63[56], which is contrary to the inactivation of p63 upon co-aggregation with mutant p53. In contrast to mutant p53, binding of Δ133p53β to p63 isoforms disrupt Δ133p53β aggregates and stimulate its oncogenic features. Overall, our study reveals that Δ133p53β aggregates are biochemically and functionally distinct from mutant p53 aggregates. Conversely, overexpression of TAp63 although similarly induced loss of the Δ133p53β aggregates resulted in reduced invasiveness of MDA-MB-231 D3H2LN. This is in agreement with published data suggesting opposite roles of TAp63 and ΔNp63 proteins in cancer progression[40].

Finally, interaction with ΔNp63α protein releases Δ133p53β from the aggregates allowing an increase of Δ133p53β intracellular mobility. This observation is fully compatible with the presence of Δ133p53β in the aggregates which freeze the mobility of this p53 isoform. Recently it was demonstrated that mutant p53 undergo liquid condensation and solid-like phase transition prior to aggregate formation[57,58]. FRAP analysis was used to distinguish between phase-separated liquid and gel or solid-like droplets which are detected by either fast or slow fluorescence recovery respectively. Comparison of their recovery dynamics demonstrates the liquid characteristics of WT p53, which contrasts with a typical solid-like phase transition for mutant p53.

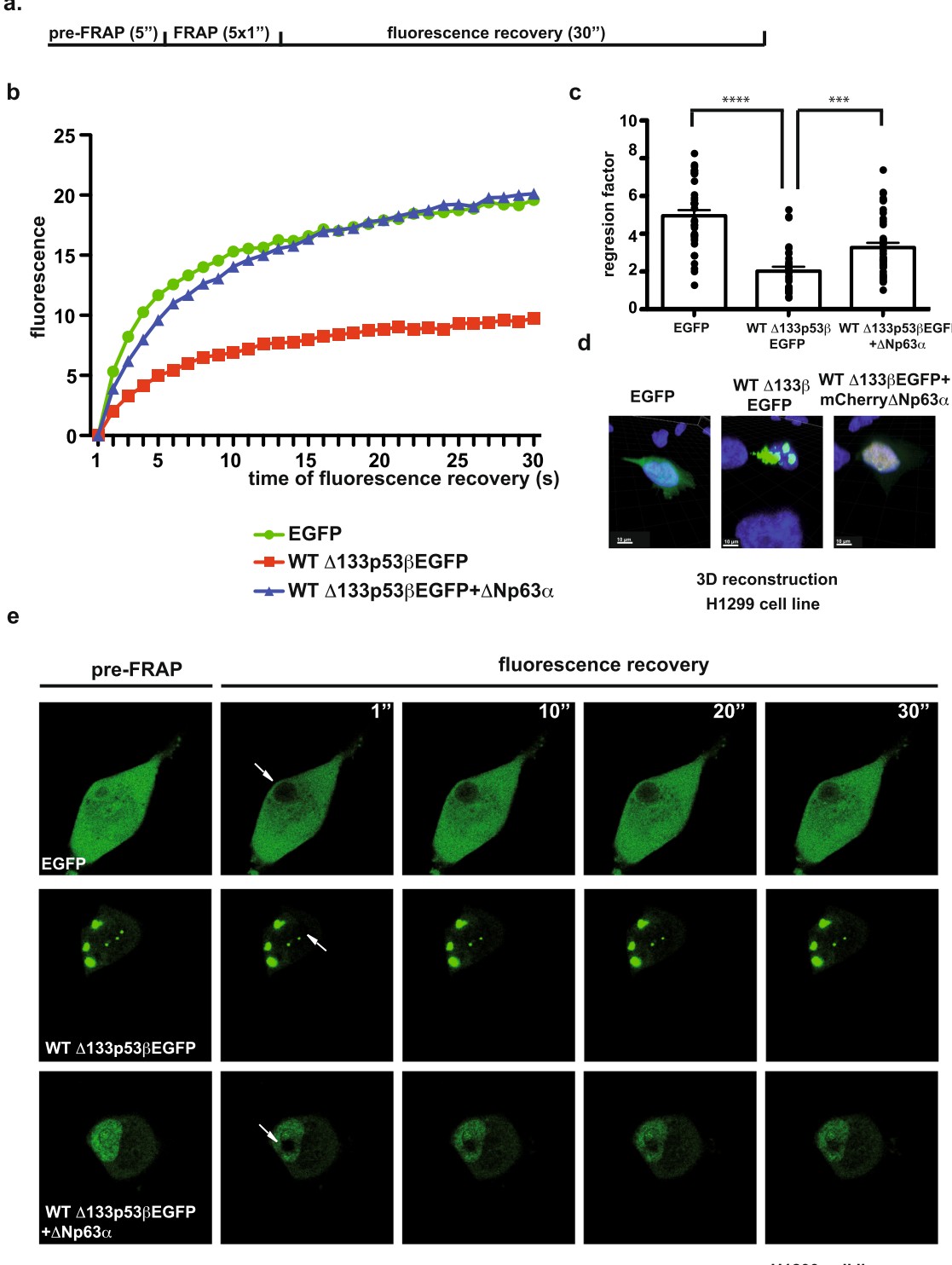

**Fig. 6 Δ133p53β molecules dynamics is highly increased in the presence of the ΔNp63α protein partner. a** Flow chart of the FRAP experiment. **b** Graphical representation of the normalised fluorescence recovery curves depleted for background fluorescence pooled from three independent experiments. SEM not shown for clarity. **c** Quantification of the regression factor with $R^2$ with no less of 0.95 pooled from three independent experiments. Values plotted are the mean of ±SEM (error bars). Statistical analysis was performed using non-parametric Mann–Whitney $t$ test with the Prism software (GraphPad). The first $p$ value is <0.0001 and second is 0.0003. **d** Immunofluorescent analysis of EGFP, WT Δ133p53βEGFP and co-expressed WT Δ133p53βEGFP and mCherryΔNp63α in H1299 cell line upon 3D reconstruction of the confocal Z-stacks. Green: EGFP (Δ133p53β), Red: mCherry (ΔNp63α), blue: Hoechst. Scale bar 10 μm. Representative images of one out of three independent experiments are shown. **e** Time-lapse images from Supplementary movies 10–12 from FRAP experiments performed on EGFP, WT Δ133p53βEGFP and co expressed WT Δ133p53βEGFP and mCherryΔNp63α in H1299 cell line. The arrows indicate FRAP zone. $n = 3$ for FRAP experiments. Source data are provided as a Source Data file.

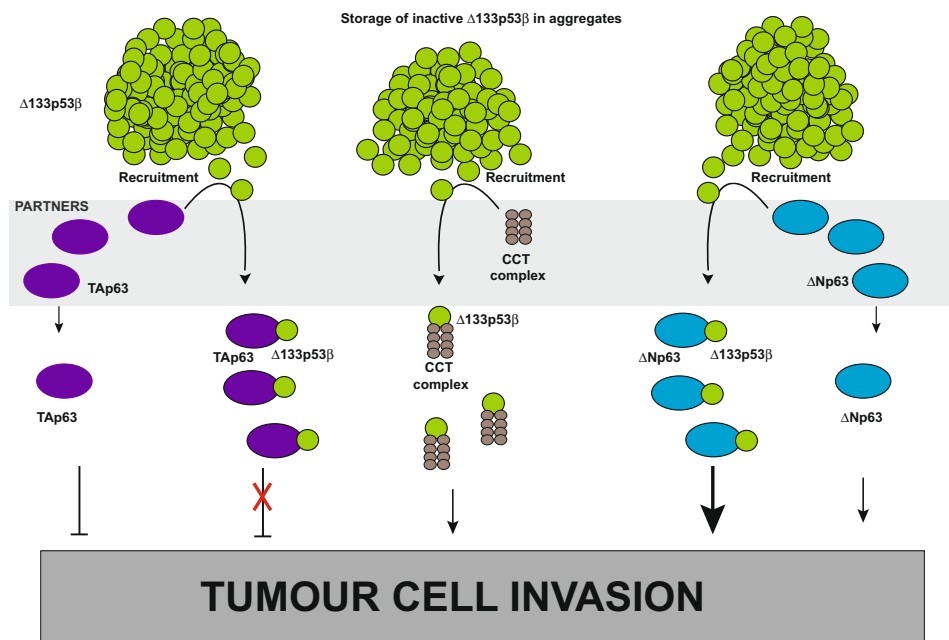

**Fig. 7 Schematic representation of the Δ133p53β activity regulation through aggregation.** Increased expression of Δ133p53β in cancer cells leads to its accumulation and creation of the aggregates. Δ133p53β can form stable aggregates mostly due to its unfolded conformation. Δ133p53β sequestration in aggregates would store isoform in inactive form. When this isoform activity is required, recruiting partners release it from aggregates. Recruiting partners are chaperones like CCT complex or p53 family members like p63 isoforms. The presence of recruiting partners increases cancer cell invasion. Interaction with CCT complex activates Δ133p53β driven invasiveness. The binding of Δ133p53β to p63 isoforms further leads to positive effects on invasion, either by synergizing the pro-invasive ΔNp63 isoforms activity or inhibiting the anti-invasive effect of TAp63 isoforms.

Similarly, we detected two distinct states of the Δ133p53β isoform, immobile in the aggregates, characteristic of a more solid-like in the absence of the interactors and mobile, liquid-like in the presence of the protein partners such ΔNp63α. These results suggest phase transition of the Δ133p53β isoform between inactive and active state in the presence of the protein partners.

Our experiments suggest a model of WT Δ133p53β activity regulation (Fig. 7). We propose that WT Δ133p53β is stored in an aggregated, inactive state. Chaperones like CCT complex are responsible to keep a basal level of the active isoform that is involved in certain cellular functions. Δ133p53β sequestration in aggregates could serve as a storage mechanism until the activity of the isoform is required. Provided that adequate protein partners like p63 family members are present, Δ133p53β is recruited from aggregates and it activates cancer cell invasion. All together this study demonstrates that cancer cell functions, like invasion, depend on balance of p53 isoforms levels, chaperone quantity and presence of specific protein partners like p63 family members. Finally, it would be very tempting to develop inhibitors of CCT and p63 interactions with Δ133p53β and evaluate their efficacy on metastasis.

## Methods
**Cell lines**. Breast cancer cell lines MCF-7, MDA-MB-231, SK-BR-3 and colon cancer SW480 were cultivated in MEM 1X (Minimum Essential Medium, Gibco ref. 10370-047), colon cancer cell lines HCT116 in McCoy's 5A (Gibco, Ref 26600-023), and lung cancer cell line H1299 in Roswell Park Memorial Institute (RPMI) 1640 Medium (Eurobio Scientific, ref. CM1RPM08). All media were supplemented with 10% FCS, glutamine and sodium pyruvate and incubated at 37 °C in the presence of 5% $CO_2$. Cells were transfected using JetPei transfection reagent (Polyplus) according to the manufacturer's instructions 24 h prior to use. Cell lines were routinely tested for mycoplasma contamination. Cells were transfected using pcDNA3-based plasmids containing different constructs (WT *Δ133TP53β*, *Δ133TP53β R175H*, R273H and R280K mutants, WT *TATP53α* and its mutants R175H and R273H) tagged with Flag, as well as *ΔNTP63α, γ* and *TATP63α*. Constructs using GFP tags were created in the pEGFP plasmid. The

mCherryΔNp63α construct was also made in pcDNA-3 plasmid. Infections were performed 24 h before use by retroviral particles produced using either pMSCV puro (or hygro) for protein expression or RNAi-Ready pSIREN-retroQ for Sh-based gene silencing (Clontech Laboratories). Sh and Si RNA sequences used in this study are listed in Supplementary Table 2. pSIREN-Luc (luciferase)-shRNA (Clontech) based viral particles was used as control.

**Antibodies**. Antibodies used in this study are listed in Supplementary Table 3. The horse-radish peroxidase (HRP)-conjugated anti-mouse (Anti mouse HRP linked antibody ref 7076 and Rabbit Anti-Mouse IgG (Light chain specific) mAb (HRP Conjugate) ref 58802) and rabbit IgG (Anti-Rabbit IgG HRP linked Antibody ref7074) antibodies were purchased from Cell Signalling Technology and were diluted at 1:5000 prior to use. HRP-conjugated anti-IgG α-goat (DAKO ref P0449) and anti-sheep (Sigma ref. A3415) were diluted 1:2500 for use. The Western Lightning Chemiluminescence (ECL) reagents were purchased from PerkinElmer. (Ref No. NEL104001EA).

Secondary, fluor conjugated anti-mouse and rabbit IgG antibodies were purchased from Thermo Fisher Scientific: Alexa fluor 555 A21424, Alexa fluor 444 A11029 and Alexa fluor 546 A11030. All antibodies were used at 1:1000 dilution.

**Protein extraction**. Protein extracts were prepared with lysis buffer (50 mM Hepes pH7.4, 150 mM NaCl, 0,5% NP40, 1 mM EDTA supplemented with protease inhibitors) for 10 min at 4 °C and centrifuged at 9200 × *g* at 4 °C for 10 min. Cleared protein extracts and pelleted cell debris were separated upon centrifugation and used for further analysis.

**Immunoblot blot analysis**. For immunoblot blot analysis cleared protein extracts or cell debris were denatured for 3 min at either 42° or 95 °C, identical protein quantity was loaded and separated on 12% SDS-PAGE prior transfer to nitro-cellulose membranes. Cell debris samples were sonicated prior denaturation. The antibodies used for immunoblot analysis and their dilutions are listed in Supplementary Table 4. All primary antibodies were incubated for 1 h room at temperature except α-Pontin which was incubated over night at 4 °C.

**Immunoprecipitation analysis**. Protein extracts were prepared as above. Cleared protein extracts were incubated with anti Flag M2 affinity gel (Sigma, Cat No A2220) or Dynabead Protein G magnetic beads (Life technologies, Cat No10003D) according to the manufacturer's instructions. Magnetic beads were used for immunoprecipitation experiments with α-p53 Ab 240 and α-p53 CM1 antibodies.

Following incubations, beads were precipitated by centrifugation or magnetic field and washed 5x with lysis buffer. After resuspension in loading buffer and 5 min incubation at 95 °C, beads were loaded on to SDS-PAGE and analysed by immunoblot.

**Immunofluorescence analysis.** For immunofluorescence analysis cells were seeded on glass cover slips, transfected with plasmids expressing indicated proteins and left for 24 h. Cells were fixed with 3.2% paraformaldehyde and permeabilised with 0.2% Triton X-100. The antibodies used for immunofluorescence analysis and their dilutions are presented in Supplementary Table 5.

Both primary and secondary antibodies were incubated for 30 min at room temperature. Nucleus coloration was obtained using Hoechst stain that was incubated alongside with secondary antibodies. After mounting with ProLong Gold anti-fade agent (Invitrogen, Molecular Probes Ref. P36930), cells were visualized using either Leica confocal TCS SP5 microscope with LAS-AF software for confocal imaging or Zeiss Axio Imager Z2 with scMOS ZYLA 5.5 camera and 40 × 1.3 NA magnification. 3D reconstructions from confocal Z-stacks were obtained using Imaris software. No fewer than 25 cells per sample in each experiment were analysed.

**Recombinant protein expression and purification.** The cDNA sequence coding Δ133p53β was prepared by IDT® gene synthesis and optimized for *E. coli* expression. This target protein was cloned in frame into pDB plasmid between NdeI and XhoI sites to give the pDB- Δ133p53β vector. In this construct Δ133p53β was fused with a (his)$_6$-MBP N-terminal tag. This tag sequence was followed by the HRV 3c (3c) protease recognition site (Leu-Glu-Val-Leu-Phe-Gln/Gly-Pro) that allows the specific cleavage between Gln and Gly, providing Δ133p53β with additional Gly-Pro at the N-terminus. The pDB- Δ133p53β plasmid was transformed in *E.coli* BL21 (DE3) and grown overnight at 25 °C in ZYM 5052 auto-induced medium supplemented with 50 µg/ml kanamycin[59]. Cells were harvested by 20 min centrifugation at $6000 \times g$ at 4 °C. The pellet was resuspended in Tris-HCl 20 mM pH 7.5, NaCl 600 mM, and 2 mM 2-mercaptoethanol (buffer A) supplemented with 1% TritonX100 and stored at −80 °C. Cells were supplemented with Complete® EDTA free tablet (Roche) and lysed by sonication. Insoluble proteins and cell debris were sedimented by centrifugation at $40,000 \times g$ at 4 °C for 30 min. The supernatant was supplemented with imidazole to 10 mM final concentration and loaded onto a 5 ml gravity affinity column (Ni sepharose 6 FF 5 ml, GE Lifescience), equilibrated with buffer B (buffer A containing 10 mM imidazole). The column was washed with 50 ml of buffer B and proteins were eluted with a one-step gradient of buffer C (buffer A containing 500 mM imidazole). The peak fractions were analysed by SDS-PAGE. Fractions containing tagged Δ133p53β were pooled and dialyzed overnight at 4 °C against buffer A. The dialyzed protein was then loaded on a Superdex S200 16/60 (HiLoad 16/600 Superdex 200 pg, GE Lifescience) equilibrated with buffer A. Peak fractions were analyzed by SDS-PAGE.

**Recombinant protein characterization.** Characterization of Δ133p53β was done by SEC and DLS. SEC was performed on a Superdex S200 16/60 (HiLoad 16/600 Superdex 200 pg, GE Lifescience) on an AKTA Pure system (GE Lifescience). The protein was concentrated and injected on the column with a 5 ml loop equilibrated with buffer A. DLS was performed using the Zetasizer NaNo-S system (Malvern Instruments) on the SEC maximum peak. Data were collected in 3 consecutive runs of 15 repeated measurements at 20 °C. The sample volume used was 150 µl at 0.8 mg/ml.

**RT-PCR.** Total RNA was extracted with the RNeasy Mini Kit (Qiagen) and treated with DNase (Qiagen) prior to reverse transcription, which was carried out using oligo(dT) (Invitrogen) and M-MLV reverse transcriptase (Invitrogen).

p63 isoforms were detected using primers listed in Supplementary Table 6 by nested PCR approach.

The following PCR program was used: 94 °C - 4 min (94 °C - 30 s, 55 °C - 30 s, 72 °C - 3 min) ×30 cycles followed by 72 °C - 8 min and 4 °C -∞

The first and second PCR reaction were performed according to the combination of the primers indicated in Supplementary Table 7.

**Immunohistochemistry analysis.** Ethical approval for work using human tumours (reference LRS/10/09/037 and MEC/08/02/061) was obtained in New Zealand and all procedures followed institutional guidelines. All individuals provided written informed consent. The clinical characteristics of the patients analysed are described in Supplementary Table 8. Tumours expressing Δ133p53β were selected upon RT-qPCR assays as described by Mehta and colleagues[60]. The RNAscope assays were performed according to Kazantseva and colleagues[24]. For immunofluorescence 4 µm sections from formalin-fixed paraffin-embedded tissues were stained using the KJC8[15] and Alexa Fluor 488 (Life Technologies) antibodies. Tissues were visualized using a Nikon C2 confocal microscope (Nikon Corporation) and NIS Elements version 4.13 imaging software.

Four µm sections of formalin-fixed paraffin-embedded tissues from Δ122p53 homozygote mice were subjected to immunofluorescence using the PAb421 antibody towards p53 (MilliporeSigma, Burlington, MA, USA), the Opal 570

fluorophore, and the automated Opal 7 colour IHC kit (Akoya Biosciences, Menlo Park, CA, USA) with 1 µg/mL Hoechst dye used instead of spectral DAPI. Slides were stained using an automated method with antigen retrieval performed using ER1 solution for 20 min (Leica Bond RX, Lecia Biosystems, Wetzlar, Germany). Stained slides were scanned into the Aperio VERSA digital pathology system at ×400 magnification (Leica Bond RX, Lecia Biosystems, Wetzlar, Germany).

**Migration and invasion assays.** H1299 cell migration was performed using a "scratch-wound assay". Confluent cells were scratched with a pipette tip and bright-field images were taken at the beginning ($T = 0$ h) and the end of the assay ($T = 12$ h) using EVOS FL microscopy system at ×4 magnification. Free space between cell front lines was quantified using ImageJ software.

MCF-7 and MDA-MB-231 D3H2LN cell invasion assays were performed as previously described[61].

**Time-lapse imaging.** Time-lapse DIC microscopy was performed on a Leica DMIRE2 or Olympus IX83 inverted microscopes with an automatic shutter and full filter sets. Captured images were converted into TIFF files, edited and compiled with Metamorph software. Images were captured every 10 min during 16 or 32 h.

**Proteomic analysis.** After immunoprecipitation, samples were loaded on a precast SDS-PAGE (12% polyacrylamide, Mini-PROTEAN® TGX™ Precast Gels, Bio-Rad). The gel was stained with the PageBlue Protein Staining Solution (Euromedex).

Enzymatic *in-gel* digestion was performed according to the Shevchenko protocol[62]. Briefly, gel slices were destained by three washes in 50% acetonitrile, 50 mM triethylammonium bicarbonate buffer and incubated overnight at 25 °C (with shaking) with 1 µg of trypsin (Gold, Promega, Madison USA) in 100 mM triethylammonium bicarbonate buffer. Tryptic peptides were extracted with 50% acetonitrile and 5% formic acid, and dehydrated in a vacuum centrifuge.

**Nano LC-MS/MS.** Peptides were solubilized in 10 µL of 0.1% formic acid—2% acetonitrile and analyzed online by nano-flow HPLC-nanoelectrospray ionization using a Q Exactive HF mass spectrometer (Thermo Fisher Scientific) coupled with an Ultimate 3000 HPLC (Thermo Fisher Scientific). Desalting and pre-concentration of samples were performed on-line on a Pepmap® precolumn (0.3 mm × 10 mm). A gradient consisting of 0–25% B for 100 min, 25–90% B for 5 min (A = 0.1% formic acid, 2% acetonitrile in water; B = 0.1% formic acid in 80 %acetonitrile) at 300 nL/min was used to elute peptides from the capillary (0.075 mm×500 mm) reverse-phase column (Pepmap®, Thermo Fisher Scientific) fitted with a stainless steel LC/MS emitter (Thermo Fisher Scientific). LC-MS/MS experiments comprised cycles of 13 events; one MS1 scan with Orbitrap mass analysis at 60,000 resolutions followed by Higher-energy C-trap dissociation (HCD) of the 12 most abundant precursors (resolution: 30.000, NCE: 28). Spectra were acquired with the instrument operating in the information-dependent acquisition mode throughout the HPLC gradient. The mass scanning range was $m/z$ 375–1500 and standard mass spectrometric conditions for all experiments were: spray voltage, 2 kV; heated capillary temperature, 270 °C; S lens RF level, 55.

All MS/MS spectra were searched for against the complete proteome set of human entries of uniprot (http://www.uniprot.org/; v 2017_06) using the MaxQuant v1.5.5.1 software[63] with default parameters. Peptides and proteins identifications were filtered using FDR of 1%.

**RNA-sequencing, correlation and gene ontology analysis.** RNA-sequencing data of 12 prostate cancer patients, previously published[25] was used for correlation and gene ontology analysis. Spearman's correlation analysis was performed to identify RNAs associated with Δ133p53 RNA expression using the cor.test() function in R with the method set to "spearman"[64]. Correlation coefficient cutoff of ±0.3 and a false discovery rate <0.05 was used to identify enriched gene sets using GSEA pre-ranked function in GSEA v4.03[65].

**FRAP analysis.** Fluorescence recovery after photobleaching (FRAP) was performed using Leica confocal TCS SP5 microscope equipped with environment control with LAS-AF software in FRAP mode. H1299 cells were transfected with next plasmids: pEGFP, pΔ133p53βEGFP and pcDNA3mCherryΔNp63α in Ibidi 35 mm glass-bottom dish (ref. 81158). Twenty-four hours post transfection FRAP experiments were performed as shown in the flow chart in Fig. 6a. Imaging was performed in 512 × 512 pixel mode. Pre bleach imaging lasted 5 s. Nuclear bleach was performed applying 5 passes of 1 s using 100% 488 nm laser power on oval ROI (region of interest) with 3 µm diameter in "zoom-in" mode. Thirty post-bleach frames were acquired in bidirectional mode. Twenty cells of each sample per experiment were imagined. Three independent experiments were performed. Recovery curves were normalised and subtracted for background fluorescence. Linear regression of fluorescence recovery curves was calculated and statistical analysis was performed for those with $R^2$ higher than 95%.

**Statistical analysis.** All data are presented as the arithmetic mean ± SEM. Statistical analysis were performed using non-parametric Mann–Whitney $t$ test with the Prism software (GraphPad).

**Reporting summary**. Further information on research design is available in the Nature Research Reporting Summary linked to this article.

## Data availability

The mass spectrometry proteomics data have been deposited to the ProteomeXchange Consortium via the PRIDE[66] partner repository with the dataset identifier PXD022998. Previously published RNA-sequencing data have been used for gene set enrichment analysis[25]. The RNA-sequencing data for 12 prostate cancers have been deposited to the NCBI Sequence Read Archive (SRA) repository (Accession number: SRP328993). Figures 1a–i, 2a–g, 3a–i, 4a–g, 5a–i, 6a–e, Supplementary Figs. 1a–c, 2a–d, 3a–p, 4a, b, 5a–f data generated in this study are provided as a Source Data file. Source data are provided with this paper.

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

## Acknowledgements
This paper was supported by grant from INCa no. 2017-169, no. CNRS: 157944 N.A., P.R. and A.K. are supported by CNRS. A.W.B., T.S., S.M. and M.K. are supported by grants from the Health Research Council and Royal Society of New Zealand. P.R. is supported by INSERM. We acknowledge the imaging facility MRI, member of the national infrastructure France-BioImaging infrastructure supported by the French National Research Agency (ANR-10-INBS-04, «Investments for the future»). We thank Véronique Gire and Philippe Fort (CRBM, CNRS, Montpellier) for helpful discussions. Mass spectrometry experiments were carried out using the facilities of the Montpellier Proteomics Platform (PPM, BioCampus Montpellier). We acknowledge Maurice Wilkins Centre for Molecular Biodiscovery for RNAseq funding.

## Author contributions
N.A. wrote the manuscript, performed experiments and analysed data. T.S., G.G., M.K., and S.M. performed experiments and analysed data. E.V., A.F., F.A., N.S., S.R., S.U., and M.S. performed experiments. A.V.K., S.U., and M.S. analysed data. J.C.B., P.B., A.V.K., A.B., and P.R. are group leaders involved in the project. T.S., M.K., S.M., T.S., and A.B. wrote the manuscript. P.R. wrote the manuscript, analysed data and supervised the study.

## Competing interests
The authors declare no competing interests.
