## [Peer Review File · Nature Communications]

Reviewers' comments:

Reviewer #1 (Remarks to the Author):

p53 is part of a family of many different isoforms and the exact regulation and function of each isoform is not known. In this manuscript the authors focus on the $\Delta 133p53\beta$ variant and determine to what extent it binds to known interactors of the WT or structural mutants of p53 and determine what the impact on aggregation, folding and invasion is. Although this is technically not known for the $\Delta 133p53\beta$ in specific, the concepts in this manuscript are not very novel as $\Delta 133p53$ has been shown invasive, it has been shown to aggregate and it is known to interact with p63. In addition, the CCT complex has previously been shown to interact with WT p53. More importantly, it is questionable to what extent these findings translate to what happens in tumours in vivo. The authors show one result of aggregation coinciding with $\Delta 133p53\beta$ expression in one breast tumour, but it is unknown if the interaction of $\Delta 133p53\beta$ with CCT or dNp63 is important in a tumourigenic model. The manuscript is well-written but some figures need a bit more clarification as detailed in the comments below.

Major comments:

1. From figure 1A the authors conclude that both the WT and the 273H do not aggregate. However, the bottom 'Flag' blot does show some aggregation for 273H albeit less than WT p53. The authors might want to moderate the statement on paragraph 3 of the results section 'but not WT or contact 273H mutant aggregates at'. More importantly in this figure, it is hard to compare the amount of aggregation one sees with the structural mutant 175H to the $\Delta 133p53\beta$ variants. It would be good to show $\Delta 133p53\beta$ with the full length versions together in at least one figure in the same blot. Same for the pellet fraction in C and D. The same holds true for the folding shown in Figure 2A and B.
2. In figure 1E and F the actual overlay with ameloid is not that easy to see. There is more yellow seen in the 175H than in the $\Delta 133p53\beta$. The arrows don't seem to point to yellow dots. Perhaps a supplemental figure to show the independent fluorescent channels for one of the images of the Z-stack would help.
3. Figure 1G should be repeated in more tumours. It would be much more informative to also know whether this aggregation occurs in non-invasive tumours only and whether this is dependent on the expression of p63 (also which isoform) or CCT expression.
4. Why is the 175H p53 band lower than 273H or WT p53 in figure 4A, but not in other figures?
5. The quality of fig 2C is poor. The authors need to give more information on which programs were used to determine the aggregation prone areas
6. The description of how the models of figure 2F were generated is limited.
7. Given that WT p53 can interact with CCT, it would be good if the authors could include a WT p53 as well as a p53 β in the interaction studies in figure 3A to compare how well $\Delta 133p53\beta$ binds to CCT in relation to these constructs.
8. Figure 3C is not mentioned in the text.
9. A role for $\Delta 133p53\beta$ in proliferation should be excluded to warrant the conclusions of figure 3H that $\Delta 133p53\beta$ promotes migration. This also applies to the invasion assays in figure 6.
10. Figure 4 does not contribute much to the manuscript and could be moved to the supplemental data.
11. Instead of figure 4, a more in-dept analysis of the role of TAp63 in $\Delta 133p53\beta$ regulation might be more relevant. In relation to this there are plenty of questions that still remain. In supplemental figure 3 it is shown that $\Delta 133p53\beta$ can also bind TAp63. In contrast to dNp63, this variant is known to inhibit migration and invasion and many researchers speculate that aggregation of mutant forms of p53 with TAp63 is preventing TAp63 from functional and so causes invasion. How does the interaction between $\Delta 133p53\beta$ and TAp63 impact invasion? MCF7 cells as used in figure 6 express TA and dNp63. To what extent is the increase in invasion due to TAp63 inhibition here? What happens in response to TAp63 overexpression or knockdown? It would be more informative if the authors would use the MDA MB231 D3H2LN line that only expresses TA as well. To what extent is the CCT complex expressed in MCF7 cells? What happens if the authors express the $\Delta 160p53\beta$ that does not bind CCT? Is binding of $\Delta 133p53\beta$ to p63 mutually exclusive for CCT or vice versa?
12. The title suggest that the authors have studied the role of $\Delta 133p53\beta$ in invasion thoroughly, but the only assays that were done in this regard are wound scratch assays in figure 3 and a couple of invasion assays in figure 6 that were done in either H1299 or in MCF7 cells. Additional validation is required, preferentially using mouse models.
13. The whole manuscript relies very heavily on overexpression. Overexpression in general can cause

aggregation. However, aggregation can be seen endogenously in some of the cell lines used. It would be good if the authors can repeat some of the key findings more endogenously.

14. The model in figure 7 is very vague and not even showing invasion.

Reviewer #2 (Remarks to the Author):

The manuscript by Nikola and coworkers describes the aggregation potential of $\Delta 133p53\beta$ and related pro-migratory effect in cancer cells. The provided data are very interesting, innovative and add new data to the current knowledge in this field. In addition, authors explored $\Delta 133p53\beta$ aggregation potential and functional activity very elegantly, using distinct cell and molecular approaches to corroborate those findings. However, there are several concerns that need to be clarified and properly reviewed by the authors. Besides, I think that the manuscript needs a careful revision to correct and improve the text. Several of these issues are listed below:

1. Concerning the general conclusion of the work that "the $\Delta 133p53\beta$ oscillates between non-aggregated, active, and aggregated, non-active states...." may not be true. The authors detect aggregates by using OC (Abeta42) that reacts with fibrillar aggregates. They proposed that WT $\Delta 133p53\beta$ aggregates are dissociated. It is quite likely that large aggregates dissociate into small oligomers that could be detected by A11 antibodies or other methods (e.g.: J Biol Chem. 2012 Aug 10;287(33):28152-62; Cell Death Differ. 2017 Oct;24(10):1784-1798; Cancer Cell. 2016 Jan 11;29(1):90-103; J Biol Chem. 2019 Mar 8;294(10):3670-3682; Chem. Sci., 2019, 10, 10802–10814; iScience. 2020 Jan 8;23(2):100820).
2. There are some abbreviations that were not specified the first time they appeared in the manuscript, such as CTT, WT, which added difficulties to comprehend the text, especially for those who are not familiar in this area.
3. I strongly recommend a revised version of the abstract, once some information per se cannot be understood without reading the complete manuscript;
4. The statement regarding $\Delta 133p53\beta$ aggregation potential as reversible should be revised, once it provides some ideas that have not been explored in this current manuscript. If the authors want to prove that the $\Delta 133p53\beta$ aggregation is reversible, in vitro solution experiments should be performed, or more sophisticated experiments should be used in cells, such as single particle fluorescence microscope (with FRAP) studies.
5. At the Introduction section, authors state that the mechanisms of action of p53 isoforms have not yet been elucidated, which is not completely true. Please update with recent published studies.
6. I strongly recommend the authors to prepare a substantial revised version of the "Methods" section, considering all these aspects:
 - Authors did not present or explain the abbreviation "WT, for WT p53 and its isoforms; please detail this at Methods;
 - some Tables have been included and not cited in the text; antibody and oligonucleotide sequences could be shown in Tables, containing data about them in a more organized format, including the experimental conditions used for these reagents. In the current version, it has been present in distinct and not standardized forms;
 - when presenting used culture media, abbreviations have been cited and not specified;
 - clarify providers and their companies;
 - the studied mutants should be more clearly detailed, and their structures could be more clearly presented;

-RNAi and Sh sequences have not been presented in a clear form; oligo sequences do not contain 5'- and -3' labels; sh sequences could be included in Tables, outside the main text;

-Protein extraction data: use "g" instead of "rpm";

-immunoblot assays: change western blot for "immunoblot"; specify total amount of proteins loaded in each immunoblot assay;

-standardize data regarding used antibodies (for immunoblot and IF assays, all used antibodies and their experimental conditions could be included in a single Table);

-Immunoprecipitation assays: control samples should use non-related IgG, but not absence of antibody, as declared in the Methods. Non-related IgG is the correct control for these experiments.

-Immunofluorescence: provide plating information for each cell line and incubation times;

- Immunohistochemistry assays: clarify how tumors and informed consent have been obtained. The sentence is confusing;

- RT-PCR assays: include oligos in a Table outside the manuscript text; describe clearly the experimental and amplification conditions ;

7. Migration assays: the authors should have used mitomycin as a proliferation inhibitor; if not, it should be clarified; without mitomycin we cannot be sure whether migrating cells are not proliferating cells;

8. Figure Legends should be improved; several key information regarding the Figures are lacking in the current provided Figure Legends;

9. Along the results description, it is not clear for all the assays which cell line is being used; these results should be more clearly presented.

10. The title of the manuscript should include somehow that the reported $\Delta 133p53\beta$ aggregation properties are related to cancer cells. In this context, once distinct experiments have been done using specific tumor cell line models, when reporting each data, these changes should be better explained in the results description. The first time authors changes lung cancer cell line H1299 to breast cancer cell lines, it has not been clearly explained the reason why it has been done. Authors only stated: "We also evaluated aggregate forming capacity of WT and mutant $\Delta 133p53\beta$ isoforms in their corresponding cellular background". However, $\Delta 133p53$ isoforms have been studied in other cancer cell lines.

11. When comparing the results regarding $\Delta 133p53\beta$ aggregation properties (data from Fig 1): authors compare aggregation properties of $\Delta 133p53\beta$ with Full length p53. However, in order to do it, analysis should have been performed in the same blot, or at least in the exact same experimental condition. It should be clarified.

12. Figures 1E and 1F: co-staining of of p53 versus OC (Abeta42) in WT and R273 and it has not been described in the text; there is difference in the IF staining patterns when comparing $\Delta 133p53\beta$ and the mutants; authors should report it; authors should inform the number of analyzed cells; in addition, it could be interesting to also evaluate the aggregates in non-tumor cells.

13. Supplementary Figure 1 & 2: Images with better resolution should be provided.

14. Page 17, phrase: "Taken together, these data clearly that WT $\Delta 133p53\beta$ has an unfolded conformation similar to p53R175H structural mutant." Some word is missing at this phrase;

15. Page 17: in some parts of the manuscript authors write CTT, in others CTT3, please uniform it.

Point-by-Point Rebuttal Letter

We are very grateful and would like to thank the Reviewers for their very careful, constructive and helpful comments. Their insightful comments have led to improved clarity and impact of this work. In response, we performed many new experiments and extended our analyses, which substantiated our conclusion. Also, the text has been clarified and toned down where appropriate.

Color code: Reviewer text in black, our answers in blue. Some newly generated data are shown as Reviewer-only Figures since they were not deemed essential for the paper. In sum, we are confident that all remaining concerns have now been adequately addressed. Added text is in RED.

The following figures were newly generated and are marked in RED.

Figure 1A
Figure 1B
Figure 1E
Figure 1G
Figure 2C
Figure 3B
Figure 5G
Figure 5H
Figure 5I
Figure 6A
Figure 6B
Figure 6C
Figure 6D
Figure 6E
Figure 7
Supplementary Figure 1C
Supplementary Figure 1D
Supplementary Figure 2C
Supplementary Figure 2D
Supplementary Figure 4A
Supplementary Figure 4B
Supplementary Figure 5D
Supplementary Figure 5E
Supplementary Figure 5F
Supplementary movie 1
Supplementary movie 2
Supplementary movie 3
Supplementary movie 10
Supplementary movie 11
Supplementary movie 12

The following figures were improved and are marked in GREEN

Figure 1C: independent fluorescent channels (new Supplementary Figure 1A) for the images of the Z-stack (previously Figure 1E)
Figure 1D: independent fluorescent channels (new Supplementary Figure 1B) the images of the Z-stack (previously Figure 1F)
Figure 1H: Improved quality
Figure 1I: Improved quality
Figure 2D: better image resolution of previous Figure 2C

The former Figure 4 is fused to the former Supplementary figure 2 and new Supplementary figures 3 is created.
Figure 5C: new experiments were added and statistical analysis were adjusted in accordance to.
Figure 5F: new experiments were added and statistical analysis were adjusted in accordance to.
The former Figure 5 is fused with the former Supplementary Figures 3B, 3C and 3D to generate the new Figure 4.

Highlights of Revision

- 1- **Extension of *in vivo* analysis: newly added data to further support *in vivo* $\Delta 133p53$ aggregates related to invasion (requested by reviewer 1). For this two different *in vivo* models were used:**
 - human cancers
 - transgenic mice
- 2- **Dynamic of aggregates mobility**
 - FRAP experiments to reveal the dynamic of $\Delta 133p53$ mobility
- 3- **Comparison of aggregation properties of $\Delta 133p53\beta$ with full length p53**
- 4- **Role of TAp63 in $\Delta 133p53\beta$ regulation**
 - newly mechanistic added data to show a more in-depth study of the TAp63 role on invasive capacities of $\Delta 133p53\beta$.

Reviewers' comments:

Reviewer #1 (Remarks to the Author): p53 is part of a family of many different isoforms and the exact regulation and function of each isoform is not known. In this manuscript the authors focus on the $\Delta 133p53\beta$ variant and determine to what extent it binds to known interactors of the WT or structural mutants of p53 and determine what the impact on aggregation, folding and invasion is. Although this is technically not known for the $\Delta 133p53\beta$ in specific, the concepts in this manuscript are not very novel as $\Delta 133p53$ has been shown invasive, it has been shown to aggregate and it is known to interact with p63. In addition, the CCT complex has previously been shown to interact with WT p53. More importantly, it is questionable to what extent these findings translate to what happens in tumours *in vivo*. The authors show one result of aggregation coinciding with $\Delta 133p53\beta$ expression in one breast tumour, but it is unknown if the interaction of $\Delta 133p53\beta$ with CCT or dNp63 is important in a tumourigenic model. The manuscript is well-written but some figures need a bit more clarification as detailed in the comments below.

Answer:

We now provide multiple examples of $\Delta 133p53\beta$ forming aggregates *in vivo*. These are from the transgenic mouse model of $\Delta 133p53$ ($\Delta 122p53$) and from several human tumors (primary triple negative breast cancer, primary ER+Pr+HER2-negative breast cancer, brain metastasis of triple negative breast cancer, brain metastasis of colorectal cancer). In addition, below in the text, the relationship between $\Delta 133p53\beta$ and p63 isoforms is explained.

Major comments:

1. From figure 1A the authors conclude that both the WT and the 273H do not aggregate. However, the bottom 'Flag' blot does show some aggregation for 273H albeit less than WT p53. The authors might want to moderate the statement on paragraph 3 of the results section 'but not WT or contact 273H mutant aggregates at'. More importantly in this figure, it is hard to compare the amount of aggregation one sees with the structural mutant 175H to the $\Delta 133p53\beta$ variants. It would be good to show $\Delta 133p53\beta$ with the full length versions together in at least one figure in the same blot. Same for the pellet fraction in C and D. The same holds true for the folding shown in Figure 2A and B.

Answer:

We agree: one cannot exclude a weak ability of mutant p53R175H to aggregate.
- We changed the text and moderated the statement on paragraph 3 of the results section: In agreement with previously reported data, structural p53R175H, but not WT created aggregates at 42°C. Contact mutant p53R273H exhibited some very low level of aggregation forming capacity (Figure 1A).

- The reviewer justifiably would like to see the comparison of the amount of aggregation observed with the structural mutant p53R175H to the $\Delta 133p53\beta$ variants. We now present a figure, which shows $\Delta 133p53\beta$ with the full length versions (WT and mutant) together in the same blot from the soluble fraction (new Figure 1A that replaces old figures 1A and B) and the pellet fraction (new Figure 1B that replaces old figures 1C and 1D) and the folding (new Figure 2C). The results show that the amount of aggregates which are created by $\Delta 133p53\beta$ are much greater than those created by mutant p53.

NOTE: The $\Delta 133p53\beta$ aggregates are much more pronounced than those formed with p53R175H. Consequently, visualization of both types of aggregates on the same blot is difficult to achieve: for the same exposure time, $\Delta 133p53\beta$ aggregates gave a saturated signal when p53R175H aggregates gave only a weak signal.

2. In figure 1E and F the actual overlay with amyloid is not that easy to see. There is more yellow seen in the 175H than in the $\Delta 133p53\beta$. The arrows don't seem to point to yellow dots. Perhaps a supplemental figure to show the independent fluorescent channels for one of the images of the Z-stack would help.

Answer:

Correct, thank you. We had accidentally moved the arrows, making the images difficult to interpret. As indicated in the text, p53R175H mutant aggregates were partially co-localised with amyloid-type aggregates (yellow dots), detected using the OC (α -A β 42) antibody (Figure 1C). Similar to p53R175H aggregates, $\Delta 133p53\beta$ only partially co-localized with amyloid-type aggregates (yellow dots) as detected with OC (α -A β 42) antibody. We did not observe a clear difference in the number and intensity of yellow dots between p53R175H and $\Delta 133p53\beta$. For both, the number of yellow dots were low that justifies the conclusion: only a partial co-localisation with amyloid-type aggregates. As requested we now present a more complete supplemental figure to see the independent fluorescent channels for the images of the Z-stack (New Supplementary figures 1A and 1B).

3. Figure 1G should be repeated in more tumours. It would be much more informative to also know whether this aggregation occurs in non-invasive tumours only and whether this is dependent on the expression of p63 (also which isoform) or CCT expression.

Answer:

Good point. Thank you.

3. Rev1: Figure 1G should be repeated in more tumours.

Answer:

We expanded our analysis and now show that aggregation of $\Delta 133p53\beta$ also occurs in several human tumors: lung, breast and colon cancers (new Table 1 and new Figure 1G). IHC using the KJC8 antibody specific for the β C-terminus of p53 isoforms was used to visualize aggregates. Aggregates were also visualized in prostate cancer in a previous publication (Kazantseva et al., Cell Death Dis. 2019 Aug 20;10(9):631. doi: 10.1038/s41419-019-1861-1., Figure 3C and Supplementary Fig. S5). In total $\Delta 133p53\beta$ aggregates were observed in four different tumour types, including primary and metastatic forms. Our data showed that $\Delta 133p53\beta$ isoform aggregates are present in an array of different primary human tumours and more markedly in matched brain metastases.

3. Rev1: It would be much more informative to also know whether this aggregation occurs in non-invasive tumours only

Answer:

Table 1 indicates the presence of aggregates in a panel of primary breast, lung and colorectal cancers and matched brain metastases. Aggregation is clearly greater in the metastases compared to the corresponding primary tumours and normal associated tissues in breast and lung cancers.

The presence of aggregates formed by $\Delta 133p53\beta$ occurs in cells that express a high level of this isoform. If visualization of the aggregates is possible, this means that $\Delta 133p53\beta$ is highly expressed and therefore most likely also present in a non-aggregated form depending on the presence of partners (p63, complex CCT) capable of recruiting the isoform from the aggregates. Therefore, our data indicate that the presence of $\Delta 133p53\beta$ aggregates is not strictly related to the invasive capacity of cells, since this depends on the presence of partners. However presence of aggregates reflects high level of $\Delta 133p53\beta$ expression.

3. Rev1: and whether this is dependent on the expression of p63 (also which isoform) or CCT expression.

Answer:

In the above mentioned publication, (Kazantseva et al., *Cell Death Dis.* 2019 Aug 20;10(9):631. doi: 10.1038/s41419-019-1861-1., Figure 3C and Supplementary Fig. S5), the $\Delta 133TP53$ RNAscope analysis revealed that the $\Delta 133TP53$ probe showed positive staining in some cells in all cancer samples which were analyzed (Fig. 3a, top left hand panel and inset). Importantly, this report indicated that the $\Delta 133TP53$ isoform expression was in tissue regions that did not stain with p63, indicating that $\Delta 133TP53$ expression, and consequently aggregates, inversely correlate with TP63 expression. This is in agreement with our data and strengthen the results showing that $\Delta 133p53\beta$ aggregates depend on the expression of p63 (both $\Delta Np63$ and TAp63 isoforms). In addition, we quantified transcript levels of $\Delta 133TP53$ isoforms using RT-qPCR and global transcriptomic expression using RNA-seq data in a cohort of patient samples already analysed (Kazantseva et al., *Cell Death and Disease*, 2019). Using data for $\Delta Np63$ target genes expression from prostate cancer TCGA dataset, we analysed a selection for genes that were inversely correlated with either $\Delta 133TP53$ or $\Delta Np63$ in prostate cancer. Using a spearman correlation cutoff of ≥ 0.3 and ≤ -0.3 genes associated with $\Delta 133TP53$ and inversely correlated with $\Delta Np63$, we identified a set of 156 genes. Gene ontology over-representation test using Pantherdb on these 156 genes (FDR cut off < 0.05) associated with $\Delta 133p53$ demonstrate that there are genes that are involved in proteasomal degradation of the proteins that are involved in protein aggregation.

Figure 1: Reviewer only: shows the inversely correlated transcripts associated with $\Delta 133TP53$ and $\Delta Np63$ and bar plot for enriched gene ontologies.

This inverse correlation between $\Delta 133TP53$ and TP63 associated transcripts were also observed at the RNA level in breast cancers in a subtype specific manner (Mehta et al, *Oncotarget.* 2018 Jun 26;9(49):29146-29161. doi: 10.18632/oncotarget.25635. eCollection 2018 Jun 26, Figure 4F). In conclusion, inverse correlation between $\Delta 133p53\beta$ and p63 in tumors in prostate and breast cancers strongly strengthen our data showing that aggregation is dependent on the expression of p63.

To further confirm this observation, we also analyzed different types of cancer aggressiveness: normal associated tissue, primary tumours and metastasis (Table 1). Aggregates were quantitated with KJC8 antibodies. Aggregates were mainly observed in primary tumours and metastasis. Aggregation tends to increase in metastasis compared to the corresponding primary tumours in breast and lung cancers (Table 1). Aggregates were not observed in lung normal associated tissues. Some rare aggregates were observed in scattered normal cells in the breast normal associated tissues, probably reflecting lymphocytes.

Table 1: Quantitation of aggregates in primary and metastatic breast, lung and colorectal cancers.

Tissue type			n	% of samples with aggregates	% of cells with aggregates
Breast tumor	Brain metastasis	Triple negative	5	100%	11, 30, 55, 70, and 77%
	Brain metastasis	Her2 positive	5	80% (n=4)	12, 15*, 19 and 25%
	Primary	Triple negative	5	60% (n=3)	18, 27*, and 30%*
	Primary	Her2 positive	5	40% (n=2)	17* and 26%
	Primary	ER+PR+ HER2-	5	20% (n=1)	Scattered positive
Breast normal associated			15	20% (n=3)	Scattered positive
Lung tumor	Brain metastasis	Adenocarcinoma	5	40% (n=2)	9* and 25%*
	Primary	Adenocarcinoma	3	0%	
Lung normal associated			3	0%	
Colorectal tumor	Brain metastasis	Adenocarcinoma	3	67% (n=2)	8* and 12%*

(primary triple negative breast cancer, primary ER+Pr+HER2-negative breast cancer, brain metastasis of triple negative breast cancer, brain metastasis of colorectal cancer).

All these additional experiments in this revision puts our findings on much firmer ground showing that aggregation is dependent on the expression of p63.

4. Why is the 175H p53 band lower than 273H or WT p53 in figure 4A, but not in other figures?

Answer:

The R175H p53 band is also slightly lower than R273H or WT p53 in figures 1A, 1B, 2A, 2C, 3B, Sup Figures 3A and 3B (old figures 4A and 4B). All the clones p53R175H, R273H or WT p53 were sequenced and they correspond to the correct sequence. We think that slight migration shift could be due to the different posttranslational modifications in cells we used.

5. The quality of fig 2C is poor. The authors need to give more information on which programs were used to determine the aggregation prone areas

Answer:

We now provide a better resolution of the image of new Figure 2D (previously fig. 2C).

The programs used to determine the aggregation prone areas were indicated in the legend of the fig 2D: "Amyloidogenic regions are predicted by using Waltz (Maurer-Stroh et al., 2010)- region 232-237 (IHNYM); PASTA (Walsh et al., 2014) and ZipperDB (Soragni et al., 2016) - region 251-257 (ILTIITL) and ArchCandy with threshold 0.370 (Ahmed et al., 2015) - region 195-270.

6. The description of how the models of figure 2F were generated is limited.

Answer:

The old figure 2F is now the new figure 2G.

The main goal of our analysis of the p53 DBD truncated model was to examine its side-chain contact map and also exposure of side-chains to the water and to compare them with the contacts observed in the complete 3D structure of the p53 DBD. For this purpose, we used the 3D structure of p53 DBD (Pdb code 1TUP), and simply truncated it using PyMol program. The contacts of the truncated model were analyzed using both PyMol (W. Delano, 2002) and CMview software (Vehlow et al., 2011).

To clarify this procedure, we added to the legend of Figure 2F (now Figure 2G) the following text:

“The truncated structure of p53 DBD was generated and analyzed by using PyMol program (J W. Delano, Pymol: an open-source molecular graphics tool, CCP4 Newsl, Protein Crystallogr. (2002) 700.)”

7. Given that WT p53 can interact with CCT, it would be good if the authors could include a WT p53 as well as a p53 β in the interaction studies in figure 3A to compare how well Δ 133p53 β binds to CCT in relation to these constructs.

Answer:

New Figure 3B contains an interaction study for WTp53 and its mutants as well as WT Δ 133p53 β and its mutants with the CCT3 subunit of the CCT complex.

p53 β was not included since we think it is beyond of the scope of this manuscript. As mentioned in the description of the figure 3B, our data indicate that the N-terminus of Δ 133p53 β is critical for binding the CCT complex. We thank the Reviewer for stimulating an interesting direction for a future study.

8. Figure 3C is not mentioned in the text.

Answer:

Thank you, we corrected it.

Figure 3C is now indicated in the Results section: « **CCT chaperone complex regulates aggregation-forming ability of WT Δ 133p53 β** ”, **second paragraph:** “To further investigate the involvement of the CCT complex in Δ 133p53 β aggregation, we depleted 3 subunits of the complex, CCT3, CCT5 and CCT7 (figure 3C).”

9. A role for Δ 133p53 β in proliferation should be excluded to warrant the conclusions of figure 3H that Δ 133p53 β promotes migration. This also applies to the invasion assays in figure 6.

Same remark as Reviewer 2, item 7:

Figure 2 -Reviewer only: analysis of the WT Δ 133p53 β invasive capacities in H1299 cell in the presence of the mitomycin.

As requested by the reviewers, we compared the migration of H1299 cells in the presence or absence of the mitomycin. H1299 cell migration was performed using a “scratch-wound assay”. Confluent cells expressing WT $\Delta 133p53\beta$ or controls treated (or not) by mitomycin were scratched with a pipette tip and bright field images were taken at the beginning (T=0h) and the end of the assay (T=24h) using EVOS FL microscopy system at 4x magnification. Free space between cell front lines was quantified using ImageJ software. As shown in panel A for quantification or panel B for images of the wound healing assay, no significant difference of the migration in the presence or absence of the mitomycin were observed.

10. Figure 4 does not contribute much to the manuscript and could be moved to the supplemental data.

Answer: see next item

11. Instead of figure 4, a more in-dept analysis of the role of TAp63 in $\Delta 133p53\beta$ regulation might be more relevant. In relation to this there are plenty of questions that still remain. In supplemental figure 3 it is shown that $\Delta 133p53\beta$ can also bind TAp63. In contrast to $\Delta Np63$, this variant is known to inhibit migration and invasion and many researchers speculate that aggregation of mutant forms of p53 with TAp63 is preventing TAp63 from functional and so causes invasion. How does the interaction between $\Delta 133p53\beta$ and TAp63 impact invasion? MCF7 cells as used in figure 6 express TA and $\Delta Np63$. To what extent is the increase in invasion due to TAp63 inhibition here? What happens in response to TAp63 overexpression or knockdown? It would be more informative if the authors would use the MDA MB231 D3H2LN line that only expresses TA as well. To what extent is the CCT complex expressed in MCF7 cells? What happens if the authors express the $\Delta 160p53\beta$ that does not bind CCT? Is binding of $\Delta 133p53\beta$ to p63 mutually exclusive for CCT or vice versa?

These issues were treated separately as follows:

11. Rev 1: Instead of figure 4, a more in-dept analysis of the role of TAp63 in $\Delta 133p53\beta$ regulation might be more relevant.

Answer:

Figure 4 (first manuscript) has been moved to the supplementary data and is the subject of the new Supplementary figure 3. We think that it is important to show that not all chaperone interactors of $\Delta 133p53\beta$ are able to recruit this isoform from the aggregates so we kept these data in the revised version as a supplementary figure. Among the chaperone proteins, only the CCT complex allows this recruitment. Figure 4 in the revised version demonstrates an interaction of $\Delta 133p53\beta$ with $\Delta Np63$. This interaction affects aggregation (panels 4D-G) as well as interaction with TA63 (panels 4H-J).

A new Figure 5 now shows (in addition to the effects of the $\Delta Np63$ on MCF-7 cell invasion capacities, which was previous Figure 6) a more in-depth study of the role of TAp63 on the invasive capacities of MCF-7 and MDA cell lines as requested by reviewer (new Figure 5: panels 5G-I). To remain consistent, this study was placed after the study on $\Delta Np63$: in the new Figure 5, panels B-F for $\Delta Np63$ and panels 5G-I for TAp63. The whole new Figure 5 shows that $\Delta 133p53\beta$ and p63 isoforms ($\Delta Np63$ and TA63) cooperate to promote invasion in different breast cancer cell lines.

11. Rev 1: How does the interaction between $\Delta 133p53\beta$ and TAp63 impact invasion? MCF7 cells as used in figure 6 express TA and $\Delta Np63$. To what extent is the increase in invasion due to TAp63 inhibition here?

Answer:

TAp63 depletion, either alone or concomitantly with depletion of $\Delta 133p53$ or β isoforms did not change invasiveness, indicating that TAp63 expression does not alter the invasive activity of MCF-7 cells (Figure 5G). We attribute the lack of effect of the TAp63 and $\Delta 133p53$ depletion to the very low invasive capacity of MCF7. This is why we then investigated whether TAp63 and $\Delta 133p53\beta$ are involved in migratory capacities of highly invasive MDA-MB-231-D3H2LN cells, as suggested by the reviewer (next item).

11. Rev 1: What happens in response to TAp63 overexpression or knockdown? It would be more informative if the authors would use the MDA MB231 D3H2LN line that only expresses TA as well.

Answer:

Depletion of endogenous TAp63 isoforms by ShRNA significantly enhanced invasiveness of MDA-MB-231-D3H2LN cells, while ectopic expression of TAp63 α decreased cell invasion, indicating that invasiveness of these cells is dependent on endogenous TAp63 (Figures 5I and 5H).

To explore for TAp63 further, we studied the simultaneous depletion of endogenous Δ 133p53 isoform and endogenous TAp63 isoform. The decreased invasiveness conferred by depletion of endogenous Δ 133p53 isoforms was not further affected by depletion of endogenous TAp63 isoform. However, depletion of endogenous Δ 133p53 isoform drastically reduced the increased invasiveness conferred by depletion of endogenous TAp63 isoforms alone (Figure 5I). Taken together these data indicate that Δ 133p53-driven invasiveness depends on endogenous TAp63 protein isoforms in triple negative MDA-MB-231 D3H2LN cells.

11. Rev 1: To what extent is the CCT complex expressed in MCF7 cells?

Answer:

We compared the expression of several CCT complex members in MCF7 and MDA-MB-231-D3H2LN. MDA-MB-231-D3H2LN cells have increased levels of Δ 133p53 β isoforms and CCT3, CCT5 and CCT7 subunits compared to MCF7 cells (new Supplementary figure 4B).

11. Rev 1: What happens if the authors express the Δ 160p53 β that does not bind CCT?

Answer:

As indicated during the description of Figure 3B, Δ 160p53 β was found to be poorly bound to the CCT3 subunit. We think that the heart of the message of this article is to focus on the Δ 133p53 β isoform aggregation capacities, as well as pro-invasive functions that heavily depend on the interaction with CCT complex. Since Δ 160p53 β is poorly interacting with CCT complex any role this isoform might be difficult to interpret. However we cannot exclude other functions of Δ 160p53 β isoform in cancer cell physiology, which could be interesting subject for some other study.

11. Rev 1: Is binding of Δ 133p53 β to p63 mutually exclusive for CCT or vice versa?

Answer:

In the new Figure 4D, we show that: when co-expressed in H1299 cell line, Δ 133p53 β and Δ Np63a bind to each other. CCT3 protein also associates with this complex as indicated. It clearly shows that when Δ 133p53 β is immunoprecipitated, Δ Np63a and CCT3 are together precipitated, indicating that their binding to Δ 133p53 β are not exclusive.

12. The title suggest that the authors have studied the role of Δ 133p53 β in invasion thoroughly, but the only assays that were done in this regard are wound scratch assays in figure 3 and a couple of invasion assays in figure 6 that were done in either H1299 or in MCF7 cells. Additional validation is required, preferentially using mouse models.

Answer:

We now present new data which link Δ 133p53 β aggregates and invasion/metastasis. For this, two different *in vivo* models were used:

- human tumours
- transgenic mice

1/ Human tumours

As we already mentioned (please see: response to reviewer 1, item 3), we analyzed the Δ 133p53 β dependent aggregation in different tumors including primary and metastatic forms of these tumors.

To summarize our data (already described above), we analyzed different types of cancer aggressiveness: normal associated tissues, primary tumours and matched metastases (Table 1). Quantification of aggregates showed that aggregation increased in the metastases compared to the corresponding primary tumours and normal associated tissues in breast and lung cancers. This increase in aggregates reflects higher level of Δ 133p53 β expression. In addition from Table 1, it is evident that more aggressive breast cancer (Triple negative vs ER+PR+ HER2-) contains more Δ 133p53 β aggregates.

2/ Transgenic mice

A well-established mouse model to study *in vivo* invasion and metastasis is the transgenic $\Delta 122p53$ model expressing an N-terminally truncated p53 mutant, $\Delta 122p53$, an analogue of human $\Delta 133p53$ isoform⁵⁻⁸. These mice are tumour prone and readily develop metastatic tumours and vascularization. Aggregates were visualized in tumour tissues from the homozygotes $\Delta 122p53$ mice (Figure 1E).

These two models indicate a clear association *in vivo* between the presence of aggregation and the extent of invasion and metastasis.

Of note, invasion assays were also conducted in the new version of the manuscript using the highly metastatic MDA-MB231 D3H2LN triple negative cell line (new Figures 5 H and I).

13. The whole manuscript relies very heavily on overexpression. Overexpression in general can cause aggregation. However, aggregation can be seen endogenously in some of the cell lines used. It would be good if the authors can repeat some of the key findings more endogenously.

Answer:

We used ectopic expression of the $\Delta 133p53\beta$ isoform in order to elucidate the mechanism of aggregation. In the new manuscript, we have added a number of experiments showing the existence of these aggregates under conditions where there is no ectopic expression. To summarize all of these results:

1. Aggregates are dependant on the endogenous $\Delta 133p53\beta$ isoform:
 - In different tumours: lung, breast and colorectal cancers (new figure 1F, new figure 1G and new Table1)
 - In a mouse transgenic model (new figure 1E)
2. These two models also indicate an association between aggregates formed by endogenously expressed isoform and invasion/metastasis.

14. The model in figure 7 is very vague and not even showing invasion.

Answer:

We present a new version of the model, which more accurately relates $\Delta 133p53\beta$ functions and its role in invasion.

Reviewer #2 (Remarks to the Author):

The manuscript by Nikola and coworkers describes the aggregation potential of $\Delta 133p53\beta$ and related pro-migratory effect in cancer cells. The provided data are very interesting, innovative and add new data to the current knowledge in this field. In addition, authors explored $\Delta 133p53\beta$ aggregation potential and functional activity very elegantly, using distinct cell and molecular approaches to corroborate those findings. However, there are several concerns that need to be clarified and properly reviewed by the authors. Besides, I think that the manuscript needs a careful revision to correct and improve the text. Several of these issues are listed below:

1. Concerning the general conclusion of the work that “the $\Delta 133p53\beta$ oscillates between non-aggregated, active, and aggregated, non-active states...” may not be true. The authors detect aggregates by using OC (Abeta42) that reacts with fibrillar aggregates. They proposed that WT $\Delta 133p53\beta$ aggregates are dissociated. It is quite likely that large aggregates dissociate into small oligomers that could be detected by A11 antibodies or other methods (e.g.: J Biol Chem. 2012 Aug 10;287(33):28152-62; Cell Death Differ. 2017 Oct;24(10):1784-1798; Cancer Cell. 2016 Jan 11;29(1):90-103; J Biol Chem. 2019 Mar 8;294(10):3670-3682; Chem. Sci., 2019, 10, 10802–10814; iScience. 2020 Jan 8;23(2):100820).

Answer:

We performed experiments using A11 antibody to detect oligomers. Supplementary figures 1C and 1D show that the labeling with the A11 antibody colocalizes neither with the aggregates, nor with the diffuse intracellular staining of $\Delta 133p53\beta$.

Moreover, the experiments of FRAP (see new figure 6) show that in its unaggregated form, $\Delta 133p53\beta$ has a high intracellular mobility, identical to that of the diffuse GFP protein used as a control.

As this antibody recognizes all types of oligomers, but not monomers or fibrils, this strongly suggests that $\Delta 133p53\beta$ aggregates and $\Delta 133p53\beta$ diffuse staining do not contain small oligomers. However we cannot formally exclude the presence of small oligomers that could be poorly detected by the A11 antibody or because of their very short half-life. Consequently the term "oscillates" has been withdrawn to take this possibility into account, but the main forms of $\Delta 133p53\beta$ are indeed the aggregated forms and the diffuse forms.

2. There are some abbreviations that were not specified the first time they appeared in the manuscript, such as CTT, WT, which added difficulties to comprehend the text, especially for those who are not familiar in this area.

Answer:

We have checked the different abbreviations used to ensure their definition when they first appear in the text.

3. I strongly recommend a revised version of the abstract, once some information per se cannot be understood without reading the complete manuscript;

Answer:

The abstract has been extensively revised for clarification and according to the new results included in this new version.

4. The statement regarding $\Delta 133p53\beta$ aggregation potential as reversible should be revised, once it provides some ideas that have not been explored in this current manuscript. If the authors want to prove that the $\Delta 133p53\beta$ aggregation is reversible, *in vitro* solution experiments should be performed, or more sophisticated experiments should be used in cells, such as single particle fluorescence microscope (with FRAP) studies.

Answer:

To further explore the reversibility of $\Delta 133p53\beta$ aggregates, we carried out Fluorescence Recovery After Photobleaching (FRAP) experiments as suggested by the reviewer (new figure 6). FRAP was used to examine the dynamics of $\Delta 133p53\beta$ in and out of aggregates. FRAP analysis of $\Delta 133p53\beta$ -EGFP alone revealed that the fluorescence within bleached aggregates recovered slowly. Conversely intracellular mobility of $\Delta 133p53\beta$ is significantly increased when $\Delta Np63\alpha$ is co-expressed.

These results confirm the high stability of the $\Delta 133p53\beta$ aggregates, as well as their capacity to be strongly reduced by $\Delta Np63\alpha$. Nevertheless, we totally agree with the reviewer: the reversibility of the phenomenon is not completely shown and is difficult to assess. We have therefore revised the notion of reversibility and replaced it with that of recruitment by $\Delta 133p53\beta$ partners, which corresponds more accurately to the results of our analysis. We completely agree and adjusted the text accordingly throughout Abstract, Results and Discussion.

5. At the Introduction section, authors state that the mechanisms of action of p53 isoforms have not yet been elucidated, which is not completely true. Please update with recent published studies.

Answer:

We thank reviewer for this observation. Actually we wanted to state that regulation of the $\Delta 133p53\beta$ isoform activity in cancer progression (not mechanism of action) is not yet elucidated. In revised version this is corrected.

“Although a body of literature shows critical roles of p53 isoforms and particularly $\Delta 133p53\beta$ in cancer progression, regulation of its activity have not been yet elucidated. Here we show that $\Delta 133p53\beta$ activity is regulated through aggregation dependent mechanism. Aggregation capacity of WT $\Delta 133p53\beta$ was evaluated applying biochemical, cellular and computational approaches as well as *in vitro* analysis of recombinant protein and confirmed by immunohistochemical detection of aggregates in mouse transgenic model as well as in human tumours.”

6. I strongly recommend the authors to prepare a substantial revised version of the “Methods” section, considering all these aspects:

- Authors did not present or explain the abbreviation “WT, for WT p53 and its isoforms; please detail this at Methods;

Answer:

We have checked the different abbreviations used to ensure their definition when they first appear in the text.

-some Tables have been included and not cited in the text; antibody and oligonucleotide sequences could be shown in Tables, containing data about them in a more organized format, including the experimental conditions used for these reagents. In the current version, it has been present in distinct and not standardized forms;

Answer:

In revised version of the manuscript data concerning antibodies, oligonucleotides and experimental conditions from Material and Method section are present in tables:

Table 2. The list of Sh and Si RNA used for used for gene silencing
Table 3. The list of the antibodies used for immunoblot and immunofluorescent analysis
Table 4. The dilutions of the antibodies used for immunoblots
Table 5. The dilutions of the antibodies used for immunofluorescent analysis
Table 6. The list of the oligos used for p63 isoforms detection by nested PCR
Table 7. The list of the oligos used for 1st and 2nd PCR reaction for p63 isoforms detection

-when presenting used culture media, abbreviations have been cited and not specified; -clarify providers and their companies;

Answer:

In revised version of the manuscript full names of the culture media are indicated as well as their producers according to the reviewer suggestions.

-the studied mutants should be more clearly detailed, and their structures could be more clearly presented;

Answer:

In revised manuscript a more detailed explanation of the p53 mutants is introduced into in the Results section, **“The WT Δ 133p53 β creates cellular aggregates”**:

“Not surprisingly, structural p53R175H, but not WT created aggregates at 42°C. Contact mutant p53R273H exhibited some very low level of aggregation forming capacity (Figure 1A). These observations are in complete agreement with literature data showing that p53 mutations that affect correct folding of the protein (structural mutants) like R175H are leading to the destructure, which is pre-requisite for aggregation. On another side mutations that affect alpha helices of the p53 protein responsible for interactions with DNA (contact mutants), like R273H do not affect correct folding and thus do not contribute much to the aggregation.”

In addition, another part of the mutants description already exists at the beginning of the “The WT Δ 133p53 β has an unfolded conformation” section of “Results” paragraph.

-RNAi and Sh sequences have not been presented in a clear form; oligo sequences do not contain 5'- and -3' labels; sh sequences could be included in Tables, outside the main text;

Answer:

In revised version of the manuscript RNAi and Sh sequences are presented without 5'- and -3' labels and are organised in Table 2 according to the reviewer suggestion.

-Protein extraction data: use “g” instead of “rpm”;

Answer:

According to the reviewer suggestion in revised version all “rpm” abbreviations are replaced by “g”.

-immunoblot assays: change western blot for “immunoblot”; specify total amount of proteins loaded in each immunoblot assay; -standardize data regarding used antibodies (for immunoblot and IF assays, all used antibodies and their experimental conditions could be included in a single Table);

Answer:

In revised version of the manuscript all above remarks of the reviewer were taken in account and introduced in the text or Tables.

-Immunoprecipitation assays: control samples should use non-related IgG, but not absence of antibody, as declared in the Methods. Non-related IgG is the correct control for these experiments.

Answer:

We note the reviewer's remark here, but the weight of evidence overall clearly shows the validity of our approach. For example, in figure 2A, immunoprecipitation with Ab240, shows immunoreactivity with p53R175H mutant only, not with WT p53 or p53R273H mutant, which is in complete agreement with previously published data, showing the validity of the approach.

-Immunofluorescence: provide plating information for each cell line and incubation times;

Answer:

In revised version of the manuscript the plating information is provided

Immunohistochemistry assays: clarify how tumors and informed consent have been obtained. The sentence is confusing;

Answer:

In the revised version of the manuscript patient consent and information as to how the tumours are obtained is rephrased. Now it states:

“Ethical approval for work using human tumours (reference LRS/10/09/037 and MEC/08/02/061) was obtained in New Zealand and all procedures followed institutional guidelines. All individuals provided written informed consent.”

RT-PCR assays: include oligos in a Table outside the manuscript text; describe clearly the experimental and amplification conditions;

Answer:

According to the reviewer's suggestion the oligos are included in the tables in the revised version of the manuscript. Table 6 contains the list of all oligos and table 7 contains combinations of the oligos used for the first and second reactions of the nested PCR for each p63 isoform. In addition, amplification conditions are indicated.

7. Migration assays: the authors should have used mitomycin as a proliferation inhibitor; if not, it should be clarified; without mitomycin we cannot be sure whether migrating cells are not proliferating cells;

Answer:

Same remark as Reviewer 1, item 9, The data show that there is no significant difference in cell migration in the presence or absence of the mitomycin.

8. Figure Legends should be improved; several key information regarding the Figures are lacking in the current provided Figure Legends;

Answer:

We have carefully checked the legends of the figures and added the necessary information for easy and understandable reading.

9. Along the results description, it is not clear for all the assays which cell line is being used; these results should be more clearly presented.

Answer:

We have now carefully the cell lines used in the biochemical assays, immunofluorescence assays, as well as in the migration and invasion assays

10. The title of the manuscript should include somehow that the reported $\Delta 133p53\beta$ aggregation properties are related to cancer cells. In this context, once distinct experiments have been done using specific tumor cell line models, when reporting each data, these changes should be better explained in the results description. The first time authors changes lung cancer cell line H1299 to breast cancer cell lines, it has not been clearly explained the reason why it has been done. Authors only stated: “We also evaluated aggregate forming capacity of WT and mutant $\Delta 133p53\beta$ isoforms in their corresponding cellular background”. However, $\Delta 133p53$ isoforms have been studied in other cancer cell lines.

Answer:

The title of the manuscript has been changed and is now: “ $\Delta 133p53\beta$ isoform pro-invasive activity is regulated through an aggregation-dependent mechanism in cancer cells”, which more accurately reflects the context of the study.

We first used the H1299 lung cancer line, because it lacks expression of p53 and p63, making manipulation by ectopic expression of these two proteins and their respective isoforms possible without any interference from endogenous p53 or p63 proteins. As mentioned now in the new version, we then demonstrated that $\Delta 133p53\beta$ -dependent aggregation is a mechanism that occurs in a wide panel of cancers including lung, breast and colorectal cancers (new Figure 1G and Table 1) as well as in tissues from the transgenic model of D133p53 (Figure 1E). To further investigate the influence of cancer aggressiveness and p53 mutational status, we then investigated $\Delta 133p53\beta$ -dependent aggregation in various breast cancer cell lines. Indeed, this cancer is well stratified into sub-types possessing different degrees of invasiveness: triple negative (represented by strongly invasive MDA-MB231), luminal B (represented by MCF7, moderately invasive), as well as different mutational states of p53, either wild type (MCF7), conformational mutant (p53R175H in SK-BR3), or contact mutant (p53R280K in MDA-MB-231). This study on breast cancer lines allowed us to extend the observation of $\Delta 133p53\beta$ -dependent aggregates and is similar to the data from our clinical study (Table 1).

11. When comparing the results regarding $\Delta 133p53\beta$ aggregation properties (data from Fig 1): authors compare aggregation properties of $\Delta 133p53\beta$ with Full length p53. However, in order to do it, analysis should have been performed in the same blot, or at least in the exact same experimental condition. It should be clarified.

Answer:

See above for reviewer 1, item 1.

12. Figures 1E and 1F: co-staining of p53 versus OC (A β 42) in WT and R273 and it has not been described in the text; there is difference in the IF staining patterns when comparing $\Delta 133p53\beta$ and the mutants; authors should report it; authors should inform the number of analyzed cells; in addition, it could be interesting to also evaluate the aggregates in non-tumor cells.

Answer:

The figures 1E and 1F are now figures 1C and 1D in the new manuscript. The co-staining of p53 versus OC (A β 42) for WT and R273 is now described in the text: WT and mutant R273H p53 proteins showed diffuse nuclear staining and almost no colocalization with OC (α -A β 42) labelled amyloid-type aggregates which is in agreement with literature data.

There is a difference in the IF staining between $\Delta 133p53\beta$ and the p53 mutants. The $\Delta 133p53\beta$ aggregates are much more pronounced than those formed with p53 mutants. This information is important and is now mentioned in the text because it confirms the difference observed by the biochemical approach (Figures 1A and 1B). We thank the reviewer for the precise observation of the images.

In the revised version of manuscript in the Immunofluorescence section of the Material and methods, information about number of analysed cells is provide - "No less than 25 cells per sample in each experiment were analysed."

Table 1 of the revised version contains data on the evaluation of the aggregates in non-tumour tissues. In normal lung tissue we were not able to detect any $\Delta 133p53\beta$ aggregation while in normal breast tissue we were able to detect some aggregates in scattered normal cells (likely lymphocytes but these were rare overall).

13. Supplementary Figure 1 & 2: Images with better resolution should be provided.

Answer:

We changed the images to provide them with better resolution.

14. Page 17, phrase: "Taken together, these data clearly that WT $\Delta 133p53\beta$ has an unfolded conformation similar to p53R175H structural mutant." Some word is missing at this phrase;

Answer:

Thank you, we corrected it: Taken together, these data clearly show that WT $\Delta 133p53\beta$ has an unfolded conformation similar to the p53R175H structural mutant.

15. Page 17: in some parts of the manuscript authors write CTT, in others CTT3, please uniform it.

Answer:

The eukaryotic group II chaperonins is hetero-oligomeric, consisting of two stacked rings of eight paralogous subunits (CCT1 to CCT8) each. In the manuscript, we indicate the wording " CCT complex" when referring to the whole complex. In some experiments, specific subunits are referred (CCT1 to CCT8). Hopefully this is now clarified in the revised version.

References:

- 1 Maurer-Stroh, S. *et al.* Exploring the sequence determinants of amyloid structure using position-specific scoring matrices. *Nature methods* **7**, 237-242, doi:10.1038/nmeth.1432 (2010).
- 2 Walsh, I., Seno, F., Tosatto, S. C. & Trovato, A. PASTA 2.0: an improved server for protein aggregation prediction. *Nucleic acids research* **42**, W301-307, doi:10.1093/nar/gku399 (2014).
- 3 Soragni, A. *et al.* A Designed Inhibitor of p53 Aggregation Rescues p53 Tumor Suppression in Ovarian Carcinomas. *Cancer cell* **29**, 90-103, doi:10.1016/j.ccell.2015.12.002 (2016).
- 4 Ahmed, A. B., Znassi, N., Chateau, M. T. & Kajava, A. V. A structure-based approach to predict predisposition to amyloidosis. *Alzheimer's & dementia : the journal of the Alzheimer's Association* **11**, 681-690, doi:10.1016/j.jalz.2014.06.007 (2015).
- 5 Campbell, H. *et al.* 133p53 isoform promotes tumour invasion and metastasis via interleukin-6 activation of JAK-STAT and RhoA-ROCK signalling. *Nature communications* **9**, 254, doi:10.1038/s41467-017-02408-0 (2018).
- 6 Kazantseva, M. *et al.* A mouse model of the Delta133p53 isoform: roles in cancer progression and inflammation. *Mammalian genome : official journal of the International Mammalian Genome Society* **29**, 831-842, doi:10.1007/s00335-018-9758-3 (2018).
- 7 Roth, I. *et al.* The Delta133p53 isoform and its mouse analogue Delta122p53 promote invasion and metastasis involving pro-inflammatory molecules interleukin-6 and CCL2. *Oncogene*, doi:10.1038/onc.2016.45 (2016).
- 8 Slatter, T. L. *et al.* Hyperproliferation, cancer, and inflammation in mice expressing a Delta133p53-like isoform. *Blood* **117**, 5166-5177, doi:10.1182/blood-2010-11-321851 (2011).

REVIEWERS' COMMENTS

Reviewer #1 (Remarks to the Author):

The authors have done an impressive amount of work for the revisions and substantially increased the quality of the manuscript. Adding more data from human tumour samples using the KJC antibody addresses the concerns regarding the relevance of the findings in vivo and more importantly suggest a correlation with invasive behaviour. Figures 4 and 5 now clearly show that d133 p53 beta binds both TAp63 and dNp63, which promotes dNp63's role in invasion and abolishes TAp63's inhibitory role in invasion although likely not through aggregation. Figure 6 is a lovely addition, showing that the aggregates formed by d133p53beta are not mobile.

My only comment is with regards to the gene set enrichment analysis in sup figure 2C and 2D. Although these data are mostly based on a previous publication, some information for the reader seems to be missing. In the gene ontology database positive and negative regulators of oligo and tetramerisation are indicated. Do the authors see a correlation with one of the two and/or is the change in expression favouring up regulation of positive regulators and/or downregulating negative ones?

Otherwise I only have noticed a few minor typos and mistakes.

1. Abstract: through 'aggregation-dependent mechanism' should be 'through an aggregation-dependent mechanism', 'Depletion of CCT' should be Depletion of the CCT'
2. The first sentence in the results section could do with a reference
3. In the results section 'Immunofluorescence analysis..... (Supplementary Figures 1C and 1D)' the authors do not mention that the d133beta with or without mutation is localised predominantly in the cytoplasm. This is mentioned later on in the manuscript, but would be useful to mention here. In this section the word 'that' after are more pronounced, should be 'than'.
4. In the section 'CCT chaperone complex regulates aggregation-forming ability of ET d133p53beta' the sentence 'In addition d133p53alpha also exhibits (Supplementary Figure 5) seems a bit out of place here and was already mentioned in the section 'The WTd133p53beta has an unfolded conformation'.
5. In the section 'Interaction with p63 family members reduces aggregation of d133p53beta' the sentence above the final paragraph references to supplementary movie 2, but this should be supplementary movie 3. This is a really nice addition to the manuscript!
6. The Y-axis of figure 3G says 'remaining surface to close'. I presume the authors mean % of wounded area at t=12 hrs. The legend does not state a time, but I presume it is similar to H
7. Figure 6C the Y-axis says regression, which should be regression

Reviewer #2 (Remarks to the Author):

The revised manuscript by Nikola and coworkers describes the aggregation potential of $\Delta 133p53\beta$ and its related pro-migratory effect in cancer cells. The provided data are very interesting, innovative and add novelty to the current knowledge in this field. In addition, authors explored $\Delta 133p53\beta$ aggregation potential and functional activity very elegantly, using distinct cell and molecular approaches to corroborate those findings.

The authors addressed most of the questions and suggestions of the reviewers.

The new FRAP data showing that $\Delta Np63\alpha$ recruits $\Delta 133p53\beta$ isoform from the aggregates to a more liquid state are particularly interesting. This seems to show transition of p53 species between solid-like states to gel- or liquid-like states. The authors should discuss these data in the context of recent results on phase separation (PS) and phase transition (PT) of WT and mutant p53 (Chemical Science (2021) doi: 10.1039/D1SC01739J; iScience 2020 Sep 1; 23(9):101517. doi: 10.1016/j.isci.2020.101517; iScience. 2019 Feb 22; 12:342-355. doi: 10.1016/j.isci.2019.01.027).

Reviewers' comments:

Reviewer #1 (Remarks to the Author):

The authors have done an impressive amount of work for the revisions and substantially increased the quality of the manuscript. Adding more data from human tumour samples using the KJC antibody addresses the concerns regarding the relevance of the findings in vivo and more importantly suggest a correlation with invasive behaviour. Figures 4 and 5 now clearly show that $\Delta 133$ p53 beta binds both TAp63 and dNp63, which promotes dNp63's role in invasion and abolishes TAp63's inhibitory role in invasion although likely not through aggregation. Figure 6 is a lovely addition, showing that the aggregates formed by $\Delta 133$ p53beta are not mobile.

My only comment is with regards to the gene set enrichment analysis in sup figure 2C and 2D. Although these data are mostly based on a previous publication, some information for the reader seems to be missing. In the gene ontology database positive and negative regulators of oligo and tetramerisation are indicated. Do the authors see a correlation with one of the two and/or is the change in expression favouring up regulation of positive regulators and/or downregulating negative ones?

Answer:

The answer is yes. Examples include RNF135 (Ubiquitin E3 ligase), KCTD5 (Adapter for E3 ligase), FUS (RNA binding protein) and TRPM2 (involved in neurodegeneration via protein aggregation) and CRTC2 (lysosomal pathway) to be positively correlated with $\Delta 133$ p53 β expression but negative correlated with Δ Np63 in PCa samples.

In the manuscript, this correlation has already been mentioned: Section Results, “**The WT $\Delta 133$ p53 β creates cellular aggregates.**”, **paragraph starting with:** “Our previous work had demonstrated presence of the $\Delta 133$ p53 β aggregates.....”The initial text was:

Enrichment of $\Delta 133$ p53 β expression clearly correlated with cellular processes linked to homo-oligo and homo-tetramerisation confirming presence of the aggregation phenomena in prostate cancer in the presence of the $\Delta 133$ p53 β isoform (Supplementary Figures 2C and 2D).

As the reviewer points out, this study has already been published. To complete the reviewer's comment, we have now included examples of positive correlations as an indication in a table noted as Supplementary figure 2E. We have added the following comment for this figure:

Enrichment of $\Delta 133$ p53 β expression clearly correlated with cellular processes linked to homo-oligo and homo-tetramerisation confirming presence of the aggregation phenomena in prostate cancer in the presence of the $\Delta 133$ p53 β isoform (Supplementary Figures 2C and 2D). Examples of positive correlation with $\Delta 133$ p53 β include RNF135 (Ubiquitin E3 ligase), KCTD5 (Adapter for E3 ligase), FUS (RNA binding protein) and TRPM2 (involved in neurodegeneration via protein aggregation) and CRTC2 (lysosomal pathway) (Supplementary Figures 2E).

Reviewer #1:

Otherwise I only have noticed a few minor typos and mistakes.

1. Abstract: through 'aggregation-dependent mechanism' should be 'through an aggregation-dependent mechanism', 'Depletion of CCT' should be Depletion of the CCT'

Answer:

Thank you, we corrected it.

2. The first sentence in the results section could do with a reference

Answer:

Bibliographical references showing functional similarities between the small p53 isoforms, particularly WT $\Delta 133p53\beta$, and mutated p53 in cancer progression, We have now cited some of the most significant.

3. In the results section 'Immunofluorescence analysis..... (Supplementary Figures 1C and 1D)' the authors do not mention that the d133beta with or without mutation is localised predominantly in the cytoplasm. This is mentioned later on in the manuscript, but would be useful to mention here. In this section the word 'that' after are more pronounced, should be 'than'.

Answer:

We agree. To be in agreement with the paragraph which describes the presence of $\Delta 133p53\beta$ aggregates, the sentence has been changed.

New sentence:

« However, a marked punctate staining revealed $\Delta 133p53\beta$ aggregates predominantly in cytoplasm for both WT and mutant proteins.”

4. In the section 'CCT chaperone complex regulates aggregation-forming ability of ET d133p53beta' the sentence 'In addition d133p53alpha also exhibits (Supplementary Figure 5) seems a bit out of place here and was already mentioned in the section 'The WTd133p53beta has an unfolded conformation'.

Answer:

We agree. This sentence is a repetition of what was described above. We took it out

5. In the section 'Interaction with p63 family members reduces aggregation of d133p53beta' the sentence above the final paragraph references to supplementary movie 2, but this should be supplementary movie 3. This is a really nice addition to the manuscript!

Answer:

The reviewer refers to this sentence :

“no aggregation of the Δ Np63 α or TAp63 α was observed during this analysis even in conditions of overexpression”

This sentence actually refers to movies showing expression of Δ Np63 α alone (Supplementary movie 2) or in combination with WT Δ 133p53 β (Supplementary movie 3).

To avoid confusion and to make it easier to read, we have changed the end of the paragraph:

“Finally we performed time-lapse microscopy on the cells co-expressing WT Δ 133p53 β and Δ Np63 α (Supplementary movie 3) or expressing either WT Δ 133p53 β (Supplementary movie 1) or Δ Np63 α (Supplementary movie 2) alone. The presence of Δ Np63 α (fused with mCherry) reduced aggregate forming capacity of WT Δ 133p53 β . Of note, no aggregation of the Δ Np63 α or TAp63 α was observed during this analysis even in conditions of overexpression (Figure 4E and 4G, 4I panels “WB p63” and “ Δ Np63 α ”, respectively and Supplementary movies 2 and 3), suggesting that this phenomenon is a hallmark of WT Δ 133p53 β alone.”

6. The Y-axis of figure 3G says 'remaining surface to close'. I presume the authors mean % of wounded area at t=12 hrs. The legend does not state a time, but I presume it is similar to H

Answer:

Yes, than you, the accurate legend is: Wounded area (% at 12 hours) which is much more precise.

We corrected it in the figure and changed the legend accordingly.

7. Figure 6C the Y-axis says regression, which should be regression

Answer:

Thank you, we corrected it.

Reviewer #2 (Remarks to the Author):

The revised manuscript by Nikola and coworkers describes the aggregation potential of Δ 133p53 β and its related pro-migratory effect in cancer cells. The provided data are very interesting, innovative and add novelty to the current knowledge in this field. In addition, authors explored Δ 133p53 β aggregation potential and functional activity very elegantly, using distinct cell and molecular approaches to corroborate those findings.

The authors addressed most of the questions and suggestions of the reviewers.

The new FRAP data showing that $\Delta Np63\alpha$ recruits $\Delta 133p53\beta$ isoform from the aggregates to a more liquid state are particularly interesting. This seems to show transition of p53 species between solid-like states to gel- or liquid-like states. The authors should discuss these data in the context of recent results on phase separation (PS) and phase transition (PT) of WT and mutant p53 (Chemical Science (2021) doi: 10.1039/D1SC01739J; iScience 2020 Sep 1;23(9):101517. doi: 10.1016/j.isci.2020.101517; iScience. 2019 Feb 22;12:342-355. doi: 10.1016/j.isci.2019.01.027).

Answer:

This is a very good suggestion. We have added the following paragraph to the discussion, along with the associated references:

Added paragraph :

Recently it was demonstrated that mutants p53 undergo liquid condensation and solid-like phase transition prior to aggregate formation (Petronilho et al., 2021; Safari et al., 2019). FRAP analysis was used to distinguish between phase separated liquid and gel or solid-like droplets which are detected by either fast or slow fluorescence recovery respectively. Comparison of their recovery dynamics demonstrates liquid characteristic of WT p53, which contrasts with a typical solid-like phase transition for mutant p53. Similarly we detected two distinct states of the $\Delta 133p53\beta$ isoform, immobile in the aggregates, characteristic of a more solid-like in the absence of the interactors and mobile, liquid-like in the presence of the protein partners such $\Delta Np63\alpha$. These results suggest phase transition of the $\Delta 133p53\beta$ isoform between inactive, stored in aggregates and active state in the presence of the protein partners.